# The monsoon effect on energy and carbon exchange processes over a highland lake in southwest of China

## Qun Du[1], Huizhi Liu[1,2], Lujun Xu [1], Yang Liu [1,2], and Lei Wang [1]

[1] State Key Laboratory of Atmospheric Boundary Layer Physics and Atmospheric Chemistry, Institute of Atmospheric Physics,Chinese Academy of Sciences, Beijing 100029, China

[2] University of Chinese Academy of Sciences, Beijing 100029, China.

*Correspondence to:* HuiZhi Liu (huizhil@mail.iap.ac.cn)

**Abstract.** Lake Erhai is a subtropical highland shallow lake on southeast margin of Tibetan Plateau, which is influenced by both South Asian and East Asian summer monsoon. Based on four years continuous eddy covariance (EC) data over Lake Erhai, the monsoon effect on water-atmosphere exchange process is investigated by comparing the energy and $CO_2$ fluxes patterns and their main drivers during pre-monsoon (March-April), monsoon (May-October) and post-monsoon (November-December) periods. The results show the atmospheric properties display a large difference for during three different periods. There is a negative difference between water surface and air temperature ($\triangle T$) during pre-monsoon period, while a positive $\triangle T$ during post-monsoon period. The diurnal sensible heat flux (Hs) is larger during post-monsoon period, while the latent heat flux (LE) is larger during monsoon period. The monthly average Hs and heat storage ($\triangle Q$) in the lake remain negative during pre-monsoon period and the early monsoon period, and they become positive in the middle monsoon period, which indicates that the lake absorbs heat at first and releases it subsequently. LE plays a dominating role in the energy partitioning of the lake. The Bowen ratio is higher during post-monsoon period. An uptake of $CO_2$ flux is observed in middle of a day during monsoon and post-monsoon period. The $\Delta T$ is the main driver for Hs and the effect of $\Delta T$ is increased as time scales are extended from half-hourly to monthly. The wind speed has a weak effect on Hs but a strong effect on LE and $CO_2$ fluxes. Similar main drivers for Hs are found during pre-monsoon period and post-monsoon period, which is also found for $CO_2$ flux, indicating a large impact of monsoon on the heat and carbon exchange process over Lake Erhai.

# 1 Introduction

There are 304 million lakes globally and they are of significant importance in determining local weather and climate through complex physical, biochemical and biological interactions (Cole et al., 2001; Downing et al., 2006; Shao et al., 2015). Because of the substantial differences in underlying surface characteristics between lake surface and its surrounding land surface (i.e., albedo, roughness, heat capacity) (O'Donnell et al., 2010), the carbon and energy exchange processes over lakes are expected to produce a different way in responding to climate change. Lakes react rapidly to a change in the atmospheric parameters and are able to modify the surrounding atmospheric circulation (Marie-Noëlle et al., 2012). Plenty of studies on water-atmosphere carbon and energy exchange process have been reported over high-latitude water bodies (Nordbo et al., 2010; Huotari et al., 2011; Mammarella et al., 2015). However, the characteristics of water-atmosphere exchange process differ in lake size, water depth, regional climate and geographical location (Liu et al., 2009). The high-altitude lakes are exposed to more extreme meteorological conditions and more sensitive to variations in meteorological forcing (Rueda et al., 2007). The shallow lakes respond more quickly to changes in the atmospheric forcing due to smaller heat capacity (Liu et al., 2012; Zhang and Liu, 2013). Understanding the turbulent exchange process between the lake surface and atmosphere and its response to atmospheric properties is essential for improving numerical weather prediction and climate models (Dutra et al., 2010; Nordbo et al., 2010).

The change of atmospheric properties over water surface can cause large fluctuations in atmospheric forcing for lake-atmosphere interactions, and subsequently affects the turbulent exchange process (Lenters et al., 2005; Liu et al., 2011; Huotari et al., 2011; Li et al., 2015). The southeasterly wind with warm moist air masses reduced and inverted the vertical temperature difference between water surface and atmosphere to be negative over a large high-latitude saline lake (Qinghai Lake) on the northeastern Qinghai-Tibetan Plateau (QTP) in China (Li et al., 2016). The cold fronts and the meteorological properties of the air masses behind cold fronts (e.g., windy, cold and dry) significantly promoted turbulent exchange of sensible heat (Hs) and latent heat (LE) through enhanced turbulent mixing thermally and mechanically, whereas southerly winds with warm and humid air masses generally suppressed turbulent exchanges of Hs and LE over a mid-latitude large reservoir in Mississippi (Liu et al., 2009; Liu et al., 2012). In response to the changes in the weather conditions, the heat balance over a large tropical reservoir in Brazil is substantially altered, and the heat loss can be twice or threefold greater during cold front days than that during the non-cold-front days (Curtarelli et al., 2014). Consistent diurnal peaks in latent heat flux during the afternoon were observed as a result of strong dry winds coinciding with peak water surface temperatures over a small subtropical reservoir in Australia (McGloin et al., 2015). An increasing sensible heat flux over the lake retarded the cooling of lower atmosphere (below 500 m) and weaken the vertical potential temperature gradient over the lake,

while increasing wind speed and vertical wind shear further facilitated the buoyancy flux to exert higher heat convection efficiency when cold air arrived over Ngoring Lake in the Tibetan Plateau (TP) (Li et al., 2017).

The $CO_2$ emissions from lakes are traditionally measured by non-continuous or indirect methods, e.g., floating chamber (Riera et al., 1999) and boundary layer transfer techniques (Cole and Caraco, 1998). The uncertainty of floating chamber method is that the flux it measures only represents a very small area and it produces biases because the disturbances in the water-air surface (Vachon et al., 2010). The boundary layer method estimates $CO_2$ flux by the difference of $CO_2$ concentration between the water and atmosphere and the gas transfer velocity, which is traditionally parameterized only by wind speed (Cole and Caraco, 1998). However, it has been reported that different processes including convection, microwave breaking and stratification could influence the gas transfer velocity (Zappa et al., 2001; Eugster et al., 2003; Podgrajsek et al., 2014). The eddy covariance technique (EC) could provide long-term continuous measurements and the high resolution data allows examining the relation between gas exchange velocity and other meteorological variables besides wind speed (Mammarella et al., 2015). The changes in atmospheric properties could affect lake-air $CO_2$ flux. Huotari et al. (2011) reported that the $CO_2$ efflux was enhanced under persistent extra-tropical cyclone activities over high-latitude water bodies. The synoptic weather events associating with extra-tropical cyclones produced larger $CO_2$ effluxes by bringing the bottom rich $CO_2$ water to the surface through upwelling, internal wave induced mixing, and mixing by convection (Liu et al., 2016). The windy and stormy days increased 16% of the annual $CO_2$ effluxes over Ross Barnett reservoir in central Mississippi, USA (Liu et al., 2016). A 15 year long time period study found that the amount of precipitation had a large effect on dissolved organic carbon (DOC) concentrations in rivers (Pumpanen et al., 2014). The waterside convection was believed to cause the higher $CO_2$ fluxes during night compared to day (Podgrajsek et al., 2014).

Lake Erhai is a subtropical highland shallow lake on the southeast margin of Tibetan Plateau, which is influenced by both South Asian and East Asian summer monsoon. The summer monsoon induces an abrupt change in large-scale atmospheric circulation and convective activity over Asia, and carries in air mass with distinct atmospheric properties (i.e., air temperature, wind direction, relative humidity) (Li and Yanai, 1996; Lau and Yang, 1997; Zhou et al., 2012). The seasonal reversals of atmospheric properties caused by summer monsoon circulation play an important role in regulating land-atmosphere heat and water exchange process (Flohn, 1957; Hsu et al., 1999). The land-atmosphere exchange process is found closely related to the onset and retreat of summer monsoon (Zhang et al., 2012). It's reported that most of available energy was transformed into Hs before the arrival of monsoonal winds, whereas LE increased and exceeded Hs after the onset of monsoonal wind (Xu et al., 2009; Mauder et al. 2006; Li et al., 2014). Considering the substantial difference in atmospheric properties during monsoon and non-monsoon periods, the water-atmosphere carbon and energy

exchange processes are expected to display large differences. However, few studies have reported the variation of heat and carbon fluxes over lakes during different monsoonal periods, and the effect of monsoon on water-atmosphere heat and carbon is not clear.

Four years continuous EC measurements with from 2012 to 2015 have been obtained over Lake Erhai. The summer monsoon generally bursts in May and retreats in October. According to the activity of summer monsoon, three monsoon periods are defined, including pre-monsoon (March-April), monsoon (May-October) and post-monsoon (November-December) period. We hypothesize that the contrasting atmospheric properties during three different monsoon periods play an important role in modulating the turbulent exchange process over Lake Erhai. The objectives of this study are to investigate the energy and $CO_2$ exchange processes and their response to changes in atmospheric properties during different monsoon periods.

## 2   Observation site and data process
### 2.1 Site description

The Lake Erhai (25°46′ N, 100°10′ E) is located on southeast margin of Tibetan Plateau, the southwest of China (Fig. 1). The altitude of the region is about 1972 m. The lake has a length of 42.6 km from south to north, and a width ranging from 3.1 to 8.8 km from east to west, with a total area of 256.5 km$^2$. The water depth of the lake varies from 10 and 20.7 m. The water depth around the tower is about 10 m. The land surface of its surrounding area mainly consists of cropland and town. Because of the subtropical climate, no ice period occurs throughout the whole year. More than 100 rivers and streams drain into the lake, with only one outlet in the southwest (Xier River). The water level is artificially regulated, ranging between 1971.1 and 1974.1 m. Due to the fully mixing of the water, only a short stratification period occurs in the middle time of the year (Feng et al., 2015).

Lake Erhai has a subtropical highland monsoon climate, characterized by a distinct wet and dry season. During monsoon period (May-October), the area is mainly controlled by both the southwest flow from tropical depression in the bay of Bengal and southeast flow from subtropical Pacific high. The moist marine air mass brings in abundant water vapor and intensive precipitation. During non-monsoon period (November-April), as a result of the southward movement of the westerlies, the area is dominated by continental air mass mainly from desert and arid area of Arab countries, and characterized by a warm and dry season. The average annual precipitation from 1981 to 2010 is 1055 mm. Majority of the precipitation concentrates in monsoon period, with an average of 895 mm. The average precipitation is only 64 mm during pre-monsoon period, and 42 mm during post-monsoon period respectively.

### 2.2 Observation

The eddy covariance instrument is mounted on a concrete platform at a height of 2.5 m (Fig. 1). The turbulent fluxes (Hs and LE) and $CO_2$ flux are simultaneously measured with an ultrasonic anemometer (CSAT3, Campbell Scientific, Logan, UT, USA) and an open-path infrared gas analyzer (LI-7500, Licor Inc., Lin coln, NE, USA). The three components of wind and virtual air temperature are measured with an ultrasonic anemometer. The $H_2O$ and $CO_2$ concentrations are acquired by the infrared gas analyzer. The EC sensors are mounted on a pipe orienting to the prevailing wind direction (southeast), which is shown in Fig. 2. A CR500 data logger (CR500, Campbell Scientific) is applied to record the measurements with a 10 Hz sampling frequency. Water temperature at eight depths (0.05, 0.2, 0.5, 1, 2, 4, 6, and 8 m below water surface) are measured with temperature probes (model 109-L, Campbell Scientific, inc., United States) to obtain the water temperature profile, which are tied to a buoy and can change with the water level. The water surface temperature (Ts) is calculated from longwave radiation. Moreover, different micro-meteorological elements are also measured at a height of 1.5 m above the platform. Air temperature and relative humidity are also measured (HMP45C, Vaisala, Vantaa, Finland). The radiation balance components, including upward and downward shortwave radiation, as well as upward and downward longwave radiation, are respectively measured with CNR1 (CNR1, Kipp & Zonen B.V., Delft, The Netherlands). Meanwhile, the photosynthetic active radiation (PAR) is also measured with an LI-190SB quantum sensor (Campbell Sci-entific inc., United States). The wind speed and wind direction is measured with a cup anemometer (034B, Met One Instruments Inc., Grants Pass, OR, USA). The Dali National Climatic Observatory, with a distance of 15 km from the flux tower, has provided the data of precipitation.

### 2.3 Data process

The raw data is checked and spikes are discarded as a result of physical noise and instrument malfunction according to the procedures suggested by Vickers and Mahrt (1997). The data measured with the AGC (active gain control) value more than 40, which is recorded by LI-7500, is also filtered. The abnormal data points with a magnitude exceeding 3.5 times of average standard deviations are needed to remove. The collected raw 10 Hz data are processed with EddyPro software, version 4.2 (Licor Inc., 2013, United States). The double rotation method is applied to adjust the coordinate system and tilt the vertical wind speed to be zero (Kaimal and Finnigan, 1994). The 30 minutes average turbulent fluxes are calculated with block average method. The time lags between anemometric variables and gas analyzer measurements are compensated by the circular correlation procedure, which determines the time lag that maximizes the covariance of two variables, within a window of plausible time lags (Fan et al., 1990). Density corrections for LE and $CO_2$ flux are also applied with Webb-Pearman-Leuning (WPL) correction procedure (Webb et al., 1980). We evaluate the uncertainty of WPL correction on $CO_2$ flux based on the raw data from October of 2015. The daily average $CO_2$

flux with and without WPL correction is $0.91 \pm 1.95$ g C m$^{-2}$ d$^{-1}$ and $-0.25 \pm 2.69$ g C m$^{-2}$ d$^{-1}$, respectively, indicating the large effect of WPL correction.The high-pass filtering effect is also corrected to compensate the flux losses at a high frequency (Moncrieff et al., 2004). Quality checks for stability test and integral turbulent characteristic test are applied to remove the low quality fluxes (Foken et al.,

5    2004). Data quality is marked following the schemes of Mauder and Foken (2004), and the high/moderate data are retained. After data quality control, the available data for LE, Hs and $CO_2$ flux account for 54% , 66% and 55%, respectively. A 3 months data gap from September to November occurs in 2014 due to the instrument failure. More detailed information about measurements and post-processing procedures could be found in our previous studies (Liu et al., 2015).

10        The drag coefficient (the momentum bulk transfer coefficient, $C_D$), Dalton number (the heat bulk transfer coefficient, $C_H$) and Stanton number (the moisture bulk transfer coefficient, $C_E$) are determined by the bulk transfer relations, which are widely used for computing ocean-air fluxes in numerical models (Fairall et al., 2003):

$$\tau = \rho_a C_D U^2 \tag{1}$$

$$H_s = \rho_a C_a C_H U (T_s - T_a) \tag{2}$$

$$LE = \rho_a L_v C_E (U \quad q ) \tag{3}$$

$\tau$ is momentum flux (N m$^{-2}$), $\rho_a$ is air density (kg m$^{-3}$), U is wind speed, $C_a$ is the specific heat

of air (1005 J kg$^{-1}$ K$^{-1}$), $T_s$ is water surface temperature (°C), $T_a$ is air temperature (°C), $L_v$ is latent heat

of vaporization (J kg$^{-1}$), $q_s$ is specific humidity at saturation (kg kg$^{-1}$), $q_a$ is specific humidity (kg

20    kg$^{-1}$).

        The heat storage (ΔQ) in the lake is also calculated:

$$\Delta Q = \rho_w c_p \frac{\overline{\Delta T_s}}{\Delta t} z \tag{4}$$

where $T_s$ is water temperature (°C) , $\rho_w$ is the density of water (kg m$^{-3}$), $C_p$ is the specific heat of

water at constant pressure (4192 J kg$^{-1}$K$^{-1}$), $\dfrac{\overline{\Delta T_s}}{\Delta t}$ is the depth-weighted time derivative of the water

column temperature (K s$^{-1}$), and $z$ is maximum depth of measured water temperature profile (m). The

ΔQ is defined as positive when it is absorbed by the lake surface (heat is stored by the lake).

Because the flux site is close to the west bank of the lake, a footprint model (Kormann and Meixner, 2001) is applied to analyse the distribution of source area contributing to the flux. The 95% of source area contributing to flux ranges from 600 m in southeast direction and 400 m in west direction during different periods (Fig. 1). Because the flux in the west direction mainly originates from the land surface, the flux in the wind direction (225$^{o}$ to 315$^{o}$) is excluded. The flux during pre-monsoon period is more affected by land surface compared to the other two periods. Nearly 80% flux origniates from the lake surface during monsoon and post-monsoon period. Appoximately 22%, 15% and 8% of flux data is filtered based on the footprint analysis.

The data recorded during rainfall is also discarded. According to the quality control procedure presented above, around 34% of the Hs, 46% of LE and 45% of $CO_2$ fluxes are removed and the retained data are analyzed in our study. Although the gap ratio is large, it's similar to other studies over lakes (Nordbo et al., 2012; Goldbach and Kuttler, 2015; Shao et al., 2015). A long large gap between August and November of exists in 2014 due to malfunction. The time of the study is Beijing local Time (UTC+8).

## 3 Results and discussion
### 3.1 Atmospheric properties during different monsoon periods

The atmospheric properties show large differences during different monsoon periods (Fig. 3; Table 1). There is a similar diurnal course for air temperature (Ta) during different monsoon periods, but with a large difference for the magnitudes. The diurnal mean Ta is the largest during monsoon period, second during pre-monsoon period and smallest during post-monsoon period. The difference for diurnal mean Ta between monsoon period and pre-monsoon period is smaller (3.4℃) than that between monsoon period and post-monsoon period (8.1℃). There is large difference for diurnal mean water surface temperature (Ts) between monsoon period and the other two periods (around 6 ℃), but small difference between pre-monsoon and post-monsoon period (around 0.2 ℃). The difference between water surface and air temperature (△T) remains negative during most of pre-monsoon period but positive during post-monsoon period, with an average value of -1.94 ℃ and 2.65 ℃, respectively. The △T has the maximum around 8:00 and minimum around 18:00, which is opposite with the diurnal pattern of Ta and Ts.

The water-air vapor pressure difference (△e) has an opposite diurnal pattern with △T, which has the maximum around 18:00 and the minimum around 8:00. Overall, △e is relatively high during

pre-monsoon period and low during post-monsoon period, with an average value of 1.10 and 0.79 kPa, respectively. There is a larger difference for $\triangle e$ between pre-monsoon period and monsoon period during 2012 and 2013, but a larger difference between post-monsoon period and monsoon period during 2014 and 2015, attributed to the annual variation of timing distribution of precipitation. The wind speed (U) has a larger difference in daytime than nighttime during different periods. The diurnal mean U is slightly higher during pre-monsoon period than the other two periods. In general, The pre-monsoon is characterized by a higher U and $\triangle e$, while the post-monsoon period is characterized by a lower U and Ta. The Ta and Ts is the highest during monsoon period. There is a large difference for average Ts (around 6℃) between monsoon period and the other two periods, but a slight difference between pre-monsoon and post-monsoon periods. The diurnal $\triangle T$ remains negative during pre-monsoon period but positive during post-monsoon period. A weak diurnal variation of $\triangle T$ is observed during monsoon period.

The wind direction over Lake Erhai displays a typical diurnal pattern during three study periods (Fig. 4). Overall, the southeast wind and west wind is dominant in daytime and nighttime, respectively, which represents as the lake breeze and land breeze. Generally, the wind direction shifts from west to southeast in the morning (around 9:00), indicating the onset of lake breeze. The lake breeze lasts until the afternoon. Then the wind direction shifts abruptly from southeast to west around 17:00, which indicates that the lake breeze transfers into land breeze. The numerical simulation of local circulation over Lake Erhai also proved the development of lake breeze circulation during daytime and land breeze circulation during nighttime (Xu et al., 2018). The wind direction switches at the time of $\triangle T$ reaching the maximum or minimum. The alternation between lake breeze and land breeze is more obvious during pre-monsoon and post-monsoon period, attributed to a larger thermal difference between land and lake surface, compared to monsoon period. Because the region is mainly dominated by westerly belt during non-monsoon season, a strong west wind is observed in nighttime during pre-monsoon and post-monsoon periods. During monsoon period, due to the northward motion of westerly belt, the west wind becomes weak in nighttime, while the southwest and southeast flow, respectively from tropical depression in the bay of Bengal and subtropical Pacific high, dominate the region of Dali area. However, the Mountain Cang, which is located close to the western side of Lake Erhai, has obstructed the passing of southwest wind and makes the southeast wind the prevailing wind over Lake Erhai during this period. Because of the continuous southeast wind in nighttime, a weak circulation of lake-land breeze occurs during monsoon period.

The characteristic of air mass from different wind directions is examined by the bin-averaged wind speed (U), air temperature and relative humidity against wind directions (Fig. 5). The U shows a large variability against wind direction. The wind speed from southeast is the highest than that from other directions, which indicates that the lake breeze is stronger than land breeze. The southeast wind is

stronger during monsoon period, whereas the land breeze, which is from the west direction, is stronger during pre-monsoon period.

As a result of the control of the maritime atmospheric mass, the air mass during monsoon period is the warmest and wettest, which has a higher Ta and relative humidity (Rh) than that during the other two periods. The characteristic of air mass also shows variability in different wind directions. The air mass from the southeast direction is warmer than that from the west direction during both monsoon and post-monsoon period, while it's opposite during pre-monsoon period. The land surface is warmed faster during the early period and cooled faster during the latter period of the whole year compared to the lake surface, which results in a warmer air mass from the land surface during pre-monsoon period while colder one during the other two periods. The difference for Rh from between pre-monsoon and post-monsoon Period is small in southeast direction but large in west direction. The Rh of air mass from west direction is higher during post-monsoon period than pre-monsoon period, which is attributed to the intensive precipitation during monsoon period.

The atmospheric stratification and bulk transfer coefficients for Lake Erhai during different monsoon periods are also analyzed, as they are fundamental parameters for computing sensible and latent heat fluxes between water surface and air in numeric models (Fairwell et al., 2003). The atmospheric surface layer is mainly near neutral stratification during the three study periods (Fig. 6). As Lake Erhai is located in a subtropical highland area, the seasonal uniformly air temperature and the plenty of cloud have contributed to the occurrence of neutral stratification. During pre-monsoon period, the near neutral stratification accounts for as much as 85% in daytime and 92% in nighttime, respectively. Compared to pre-monsoon period, the near neutral stratification declined about 20% in daytime and 10% in nighttime during the other two periods, as a result of the increase of weakly unstable and unstable stratification. The weakly unstable stratification accounts for about 12% during monsoon period and post-monsoon period, whereas only 3% during pre-monsoon period. Most of the unstable stratification occurs during monsoon and post-monsoon periods, and the percentage is much higher in daytime (20%) than nighttime (about 5%), while it's scarcely observed during pre-monsoon period. On the contrary, the stable stratification is hardly observed during post-monsoon period. The difference for the atmospheric stability during three periods is primarily caused by the variation of $\Delta T$, which can be roughly used as an indicator of atmosphere stability (Derecki, 1981; Croley, 1989). An unstable stratification typically associates with an positive $\Delta T$ (Ts > Ta). On diurnal scale, the Ts is higher than Ta in most of the time during post-monsoon period while it's opposite during pre-monsoon period (Fig. 3), which results in the occurrence of unstable stratification during post-monsoon period and stable stratification during pre-monsoon period, respectively. The stable stratification was also observed in the spring and summer in high-latitude lakes since the Ts increases much more slowly than the overlying Ta (Oswald and Rouse, 2004). While in the fall and winter, the air temperature decreases

faster than the water surface, resulting in a positive $\triangle T$.

The relationship between wind speed and bulk transfer coefficients during different monsoon periods is shown in Fig. 7. The drag coefficient decreased fast with increasing wind speed when wind speed is lower than 8 m s$^{-1}$. When wind speed increased, the Stanton number first decreased and then gradually increased. The Dalton number changed rapidly only under a very low or high wind speed, and remained constant at most time. The negatively relationship between bulk transfer coefficients and wind speed under lower wind speed is also found in other lake studies (Yusup and Liu, 2016; Xiao et al., 2013; Verburg et al., 2010). Although there doesn't exist an obvious tendency that the bulk transfer coefficients increase with wind speed increased in our study, it's not contradictory to the bulk parameterization scheme in COARE (Fairwell et al., 2003), as the wind speed over Erhai Lake corresponds to the transitional range ($< 10$ m s$^{-1}$). The observation also indicates that a larger bias maybe caused under weak wind conditions when simulating water-air fluxes for shallow water regimes (Xiao et al., 2013). The $C_D$ during pre-monsoon and post-monsoon is close to each other and both larger than that during monsoon period. There is a relatively larger $C_H$ and lower $C_E$ during pre-monsoon period compared to the other two periods. The $C_D$ is larger than $C_H$ and $C_E$ during all periods, which is consistent with other lake study (Nordbo et al., 2011).

## 3.2 Diurnal pattern of energy balance components and CO$_2$ flux during different monsoon periods

The diurnal pattern of net radiation (Rn) is larger during pre-monsoon period, and lower during post-monsoon period (Fig. 8). The difference for maximum diurnal Rn is around 60 Wm$^{-2}$ between pre-monsoon period and monsoon period, and around 76 Wm$^{-2}$ between Monsoon period and post-monsoon period, respectively. The diurnal pattern of sensible heat flux (Hs) is consistent with $\triangle T$, which reaches the maximum in the morning and the minimum in the afternoon. The diurnal Hs is largest during post-monsoon period, with the maximum value of 28 Wm$^{-2}$ in 2013. The fluctuation of diurnal Hs is very small during Monsoon period, with a difference between the maximum and the minimum of about 10 Wm$^{-2}$, attributed to the weak diurnal variation of $\triangle T$. The diurnal Hs remains positive during monsoon and post-monsoon Period. The diurnal Hs has a greater amplitude during pre-monsoon period, with its magnitudes ranging from -24 to 5 Wm$^{-2}$. The diurnal Hs during pre-monsoon period remains negative in most time of a whole day, and changes to be positive for a short time in the morning.

The latent heat flux (LE) has an opposite diurnal pattern with Hs, which reaches the maximum in the afternoon and the minimum in the morning. The maximum diurnal LE during three study periods is close to each other, with a value around 130 Wm$^{-2}$. Diurnal LE remains at a relatively high level in most time of a whole day during monsoon period compared to the other two periods. The difference of LE between three periods is more evident in nighttime than daytime. The larger LE during monsoon period

than the other two periods, is not consistent with the variation of $\triangle$e, which is larger during pre-monsoon period, indicating a weak relation between $\triangle$e and LE. Since there is a higher Hs and lower LE during post-monsoon period, the Bowen ratio is higher during this period than the other two periods, with an average value of 0.16. There is a relatively small difference for Bowen ratio between post-monsoon and monsoon periods in nighttime, and between pre-monsoon period and monsoon period in daytime.

The diurnal pattern of storage heat in the lake ($\triangle$Q) is similar with Rn, with the maximum value occurring at the noon. The diurnal $\triangle$Q remains negative in nighttime and changes to be positive after sunrise, indicating the heat is released to the atmosphere in nighttime and absorbed to the lake in daytime. The diurnal mean $\triangle$Q during pre-monsoon, monsoon and post-monsoon period is 28.9, 5.2 and -14.5Wm$^{-2}$, respectively. The lake absorbs more heat flux in daytime and releases less in nighttime during pre-monsoon period in most years compared to the other two periods.

The diurnal pattern of LE and Hs are not in the same phase with Rn, which was also reported in highlatitude and midlatitude water bodies, and they were more closely related with other meteorological variables (i.e., $\triangle$e, $\triangle$T, U) (Assouline et al., 2008; Nordbo te al., 2001). There is a smaller difference for diurnal LE during three different periods, whereas there is a larger difference for diurnal Hs, indicating a larger effect of monsoon on heat exchange process over Lake Erhai. The LE over Lake Erhai has a large diurnal variation compared to the midlatitude reservoir (Liu et al., 2012).

It's generally acquired that the transfer way of momentum and heat are different, because the former one is dependent on the pressure gradients and the other one could transfer by molecular diffusion. In order to compare the difference between momentum and heat/water vapor roughness lengths, the Dalton numbers (Da) and Stanton numbers (St) are also calculated based on a semi-empirical relationship with the roughness Reynolds number (Brutsaert, 1975). A clear diurnal variation of Da$^{-1}$ and St$^{-1}$ is observed (Fig. 8) during three different periods. The minimum is observed at midday time and the maximum is observed at before the sunrise and after the sunset. The Da$^{-1}$ and St$^{-1}$ is lower during monsoon period, with an average of 28.8$\pm$2.5, and 28.7$\pm$2.6. The Da$^{-1}$ and St$^{-1}$ during pre-monsoon period has a larger amplitude and magnitude.

Two peaks are observed for diurnal variation of $CO_2$ fluxes during the whole study periods, one occurs in the early morning and the other one in the evening. The $CO_2$ fluxes could switch to be negative around the noon time during monsoon and post-monsoon period, indicating the $CO_2$ flux uptake in the middle of a day during these two periods. The observed maximum diurnal average $CO_2$ flux was -0.53$\pm$1.66 $u$mol m$^{-2}$ s$^{-1}$ during monsoon period (2014) and -1.62$\pm$1.52 $u$mol m$^{-2}$ s$^{-1}$ during post_monsoon period (2013), respectively. The $CO_2$ flux uptake is believed to be caused by the phytoplankton due to the eutrophication of Lake Erhai. It has been reported that the shallow lake is more affected by the rich phytoplankton (Huotari et al., 2011; Shao et al., 2015). However, the $CO_2$

uptake is weaker in daytime during pre-monsoon period compared to other periods. The seasonal fluctuation of phytoplankton in Lake Erhai has been reported by some researchers. Yu et al. (2014) observed that the concentration of Chl a and phytoplankton in Lake Erhai were higher in mid-summer and autumn and fell down from winter until April.

**3.3 Daily and Monthly average of turbulent fluxes during different monsoon periods**

The daily average Hs during pre-monsoon is lower than that during other two periods (Fig. 9). The average daily Hs during post-monsoon period has a larger annual variation compared to other two periods. The difference of daily average Hs among four years could be as large as 11 W m$^{-2}$ during post-monsoon period. The LE has a small variation among different years compared to Hs. The daily average LE is larger during monsoon period, ranging from $102.5\pm39.2$ W m$^{-2}$ to $114.8\pm29.6$ W m$^{-2}$. The $\triangle$Q is observed to have a larger annual variation. The daily average $\triangle$Q remain positive during pre-monsoon period and negative during post-monsoon period. The daily average heat absorption during pre-monsoon period is $29.8\pm13.4$ W m$^{-2}$and heat emission during post-monsoon period is -13.8 $\pm7.8$ W m$^{-2}$. Although the $CO_2$ uptake is observed during midday time, the daily average $CO_2$ fluxes remain positive during different periods. The daily average $CO_2$ flux is larger during pre-monsoon period compared to other two periods. The daily average $CO_2$ flux is $0.55\pm0.23$ g C m$^{-2}$d$^{-1}$ and $0.19\pm0.08$ g C m$^{-2}$d$^{-1}$ during pre-monsoon and post-monsoon period respectively.

The monthly average energy balance components (Rn, $\triangle$Q, LE, and Hs) show clear variations during three different periods from 2012 to 2015 (Fig. 10). Rn gradually increases from pre-monsoon period to the early monsoon period and then decreases until post-monsoon Period. The monthly average Rn during post-monsoon period is lower than 40% of the other two periods. LE remains at a higher level during three periods, with an average monthly value of 89.1, 103.9 and 82.2 Wm$^{-2}$ during pre-monsoon, Monsoon and post-monsoon periods, respectively. The average monthly LE/Rn has a smaller annual variation during pre-monsoon period, which ranges from 0.50 to 0.82 for four-year study period. LE/Rn has a larger fluctuation during monsoon period, which varies from 0.53 to 1 from 2012 to 2015. As a result of the decrease of Rn, LE/Rn exceeds 1 rapidly during post-monsoon period, with an average value of 1.68. LE plays a dominating role in energy partitioning of the lake, while Hs dominated the major proportion of Rn in the terrestrial land surface (Roth et al., 2017). The monthly average Hs remains negative during pre-monsoon period and the early monsoon period, and switches to be positive in the middle monsoon period. The magnitude of monthly average Hs remains at a very low level, with a value of less than 14 Wm$^{-2}$ during three periods. There is a negative monthly average Hs/Rn during pre-monsoon due to the negative monthly Hs, indicating the Rn is consumed by heating the water body. During the monsoon period, the Hs/Rn is still very low with an average value of 0.06, and the Rn is primarily used for evaporation. Correspondingly, a positive monthly average $\triangle$Q is

observed during pre-monsoon and early monsoon period, and a negative monthly average △Q during the other periods, indicating the lake absorbs heat at first and releases it subsequently. The average monthly △Q during pre-monsoon and post-monsoon period is 28 and -14 Wm$^{-2}$, respectively. The Hs/Rn increases to 0.20 and (Hs+LE)/Rn reaches as high as 1.72 during post-monsoon period. The excessive portion of the energy released from the heat storage in the lake is transferred through the turbulent exchanges, which is similar with other lakes (Rouse et al., 2005). A lower Hs/Rn with an annual average of 0.16 and a higher LE/Rn with an annual average of 0.81 was also observed in a midlatitude reservoir (Liu et al., 2012). However, other lakes have been reported to have a positive Hs on monthly scale over a whole year (Liu et al., 2012; Li et al., 2015). The difference between them and Lake Erhai is likely to be attributed to the positive △T and unstable stratification throughout the year in this midlatude reservoir whereas a negative △T during pre-monsoon Period and the majority portion of near neutral stratification over Lake Erhai.

### 3.4 Main drivers for Hs during different monsoon periods

The correlation coefficients between Hs and meteorological variables (ΔT, U, Ta, Ts, Rn, and Rain) on half-hourly, daily and monthly scales are investigated during three different periods (Table 2). On half-hourly scale, the product of U and ΔT (U·ΔT) is the main controlling factor for Hs, which could explain 44% and 30% variance of Hs during monsoon and post-monsoon period, respectively. However, during pre-monsoon period, both ΔT and the product of U and ΔT are found to have a major effect on half-hourly Hs. A close relationship between half-hourly ΔT and Hs is also observed during monsoon period, with a Pearson correlation coefficient between them of 0.56. On daily scale, the ΔT is found to be most closely related with Hs during all three different periods, with a correlation coefficient ranging from 0.55 during post-monsoon period to 0.78 during monsoon period. Besides, Rn is also one of the major drivers for Hs during monsoon period, as the rain mainly falling during monsoon period, the role of Rn on Hs becomes noticeable. The product of U and ΔT is observed to have a main effect on Hs only during pre-monsoon period. The Ta is also responsible for the variation of daily Hs during pre-monsoon and monsoon periods, which explain about 30% variance of daily Hs. The factors controlling Hs during monsoon period are similar with that during post-monsoon period on monthly scale. The ΔT remains as the most significant factor controlling monthly Hs, which could explain 85% and 89% variance of Hs during monsoon period and post-monsoon period, respectively. The product of U and ΔT also shows close relationship with monthly Hs during monsoon and post-monsoon periods. The similar relationship between meteorological variables and monthly Hs is not observed during pre-monsoon period. Only the rain and Rn are found to be closely correlated with monthly Hs during pre-monsoon period. The correlation between rain and Rn is also observed during monsoon period, indicating the large effect of Rn during these two periods.

In general, no significant relationship between U and Hs is observed from half-hourly to monthly scales during all three study periods. The $\Delta T$ remains as the main factor controlling Hs on all temporal scales during three periods, and the effect of $\Delta T$ is increased as the correlation coefficients between them rises when time scales are extended from half-hourly to monthly. During monsoon period, the role of Rn on Hs becomes more important. The relationship between meteorological factors and Hs is more complicated during monsoon period, while there are similar main drivers for Hs during pre-monsoon period and post-monsoon period.

### 3.5 Main drivers for LE during different monsoon periods

The relationship between LE and meteorological variables during pre-monsoon, monsoon and post-monsoon periods from half-hourly, daily and monthly scales from 2012 to 2015 are also analyzed (Table 3). Unlike the relationship between U and Hs, a significant relationship between U and LE is observed on all temporal scales during three different periods, with a higher correlation coefficient ranging from 0.51 to 0.80. However, the range of the correlation coefficients between U and LE is similar on different time scales, indicating that the effect of U doesn't increase as time scale changes. The large effect of U on LE has been reported in small lakes. The product of U and $\triangle e$ is the second major factor controlling LE during three periods, especially on half-hourly and daily scales, with a correlation coefficient ranging from 0.48 to 0.71. On monthly scale, the significant effect of the product of U and $\triangle e$ on LE is only observed during pre-monsoon period, which could explain 60% variance of monthly LE. During monsoon period, both Ta and Rn show a close relationship with LE on monthly scale. The effect of Rn on LE has also been reported in other studies. Because the variation of monthly Ta is mainly determined by the magnitude of the available energy, the close relationship between Ta and monthly LE reflects the effect of Rn on LE during monsoon period. During monsoon period, Rn is also found to be responsible for variation of LE on monthly scale, which is similar with monthly Hs. The close relationship between Rn and LE was also observed in a small boreal lake (Nordbo et al., 2001; Goldbach and Kuttler, 2015).

### 3.6 Main drivers for $CO_2$ flux during different monsoon periods

The correlation coefficients between $CO_2$ flux and meteorological variables during three different periods from half-hourly to monthly scales are shown in Table 4. On half-hourly scale, there is a similar relationship between meteorological variables and $CO_2$ flux during pre-monsoon and post-monsoon period. Both Rn and photosynthetic active radiation (PAR) are found to have a slight higher correlation coefficient with $CO_2$ flux than other meteorological variables, which indicates that the carbon exchange process over Lake Erhai is both affected by physical and biological process. The U is found to have a significant relationship with $CO_2$ flux during monsoon period. There is a relatively high correlation

coefficient between U and $CO_2$ flux on all temporal scales during monsoon period, which increases from half-hourly (0.23) to monthly scales (0.81). During post-monsoon period, the U also has a large impact on $CO_2$ flux mainly on longer temporal scale. The correlation coefficient between them is also the highest on daily and monthly scales during post-monsoon period. The U could mediate the vertical transport of gases by producing turbulent eddies across the air-water interface (Eugster et al., 2003).

On monthly scale, the major drivers of $CO_2$ flux vary greatly during different periods. PAR is found to be the most significant driver for monthly $CO_2$ flux during pre-monsoon period. During monsoon period, the main drivers for monthly $CO_2$ flux (i.e., Rn, Ta and U) are in good accordance with LE. The relationship between Ts and $CO_2$ flux is likely to be attributed to that the variation of Ts could influence the solubility in the water (Shao et al., 2015). The $\triangle$e also has a large effect on monthly $CO_2$ flux during both pre-monsoon and post-monsoon periods. A negative correlation coefficient between rain and $CO_2$ flux is observed on daily and monthly scale during pre-monsoon period. It has been reported that the more rain could bring more nutrients in to the water body, which ultimately promoted the photosynthesis of the phytoplankton (Shao *et al.*, 2015). However, a positive correlation coefficient between rain and $CO_2$ flux is also observed on monthly scale during post-monsoon period. The rain could also promote $CO_2$ emission by enhancing the transport of carbon from land/catchment areas to the water system (lateral fluxes), which enhanced the DOC and potentially the $pCO_2$ in the water (Pumpanen et al., 2014).

Overall, because the correlation coefficients between meteorological variables and $CO_2$ flux are comparatively lower on half-hourly scale, large uncertainties exist for the main drivers controlling half-hourly $CO_2$ flux. The main drivers of $CO_2$ flux are more complicated on monthly scale during three periods, indicating multiple factors may govern $CO_2$ exchange process, including biological and physical process (Vesala et al., 2006). More similar controlling factors for $CO_2$ flux are observed during pre-monsoon and post-monsoon periods, compared with monsoon period, indicating the large impact of monsoon on carbon exchange process.

## 4   Conclusions

Lake Erhai is a subtropical shallow lake located in the key regions of water-vapor transportation passages, which is influenced by both South Asian and East Asian summer monsoon. The contrasting atmospheric properties during monsoon and non-monsoon periods provide an excellent opportunity to examine the effect of monsoon on turbulent exchange process over the lake surface. The pre-monsoon period is characterized by a higher U and $\triangle$e, with a negative $\triangle$T, while the post-monsoon period is characterized by a lower U and Ta, with a positive $\triangle$T. The air mass during monsoon period is the warmest and wettest. The monsoon period has a much higher Ts than the other two periods. The southeastly wind and westly wind is dominant in daytime and nighttime, respectively. The lake and land

breeze circulation is stronger during pre-monsoon and post-monsoon period. The near neutral stratification occupies the major proportion during the three study periods. The negative diurnal $\triangle T$ during pre-monsoon period has contributed to the occurrence of stable stratification, while the positive $\triangle T$ has contributed to the occurrence of unstable stratification during monsoon and post-monsoon periods. The monsoon also has an effect on bulk transfer coefficients. The drag coefficient during monsoon period is lower compared to the other two periods. The Dalton number is larger but the Stanton number is lower during pre-monsoon period than that during other two periods.

Due to the effect of cloud, the Rn during monsoon period is lower than pre-monsoon period. The albedo is higher during post-monsoon period but similar between pre-monsoon and monsoon periods. The diurnal pattern of $\triangle Q$ is consistent with Rn, while diurnal Hs and LE are out phase of Rn, which are consistent with $\triangle T$ and $\triangle e$, respectively. The diurnal Hs during pre-monsoon period remains negative in most time of a whole day, and changes to be positive for a short time in the morning. Diurnal LE remains relatively high most time of a whole day during monsoon period. The higher Hs and lower LE has resulted in a higher Bowen ratio during post-monsoon period. LE dominates the energy partitioning of the lake, and LE/Rn exceeds 1 during post-monsoon period due to the rapid decrease of Rn. The monthly average $\triangle Q$ is positive during pre-monsoon and early monsoon period, and becomes negative during the other periods, indicating the lake absorbs heat at first and then releases it.

The $\Delta T$ remains as the main factor controlling Hs on all temporal scales during three periods, and the effect of $\Delta T$ is increased when time scales are extended from half-hourly to monthly. The factors controlling LE are more consistent from half-hourly to monthly scales compared to Hs. Significant relationship between U and LE is observed on all temporal scales, and the product of U and $\triangle e$ is the second major factor controlling LE during three periods. The U is also found to have a strong relationship with $CO_2$ flux during monsoon period. The high wind is expected to cause variations in the mechanical mixing and thus modulate the water surface turbulent and $CO_2$ exchange process (Zhang and Liu, 2013). During monsoon period, the Rn plays an important role on monthly variation of Hs and LE. Compared with monsoon period, similar main drivers for Hs are found during pre-monsoon period and post-monsoon period, which is also found for $CO_2$ flux, indicating the large impact of monsoon on heat and carbon exchange process over Lake Erhai. On monthly scale, the main drivers of $CO_2$ flux are more complicated during three periods. In future, more studies are needed to investigate the combing effect of biological and physical process on carbon exchange process over highland shallow lake.

*Acknowledgements.* This work was supported by National Natural Science Foundation of China (No. 91537212, No. 41661144018 and No. 41505007) and the National Key Research and Development Program of China (2017YFC1502101). We thank the related persons in Yunnan Provincial Institute of

Meteorology and Dali National Climatic Observatory for their maintaining the site and providing historical data.

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

**Table 1**. Daily average air temperature (Ta), water surface temperature (Ts), difference between water surface temperature and air temperature ($\Delta T$), vapor pressure difference ($\triangle e$) between water surface (es) and the air (ea), wind speed (U), the downward and upward shortwave radiation flux (Rs_downwell and Rs_upwell), the downward and upward long wave radiation flux (Rl_downwell and Rl_upwell), and albedo during pre-monsoon, monsoon and post-monsoon period from 2012 to 2015.

| Period | Year | Ta | Ts | $\triangle T$ | $\triangle e$ | U | Rs_down | Rs_up | Rl_down | Rl_up | Albedo |
|---|---|---|---|---|---|---|---|---|---|---|---|
| pre-monsoon | 2012 | 16.02 | 14.24 | -2.42 | 1.07 | 3.20 | 202.40 | 13.43 | 319.94 | 384.92 | 0.07 |
| | 2013 | 16.92 | 15.36 | -1.56 | 1.20 | 2.89 | 230.60 | 14.83 | 313.91 | 390.47 | 0.07 |
| | 2014 | 17.14 | 15.04 | -2.09 | 1.21 | 2.93 | 239.03 | 14.37 | 310.49 | 388.87 | 0.06 |
| | 2015 | 16.95 | 15.29 | -1.70 | 0.94 | 3.13 | 228.29 | 15.12 | 316.93 | 390.30 | 0.07 |
| | Average | 16.76 | 14.98 | -1.94 | 1.10 | 3.04 | 225.08 | 14.44 | 315.32 | 388.64 | 0.07 |
| monsoon | 2012 | 19.81 | 20.92 | 1.09 | 0.75 | 2.81 | 201.62 | 12.55 | 366.44 | 420.68 | 0.06 |
| | 2013 | 19.52 | 20.93 | 1.41 | 0.74 | 2.67 | 201.90 | 12.42 | 368.08 | 421.01 | 0.06 |
| | 2014 | 21.06 | 21.19 | 0.12 | 0.94 | 2.81 | 223.74 | 11.88 | 374.06 | 424.08 | 0.06 |
| | 2015 | 20.23 | 21.13 | 1.22 | 0.98 | 3.16 | 212.20 | 13.45 | 365.82 | 422.05 | 0.07 |
| | Average | 20.15 | 21.04 | 0.96 | 0.85 | 2.86 | 209.87 | 12.57 | 368.60 | 421.95 | 0.06 |
| post-monsoon | 2012 | 13.04 | 15.17 | 2.13 | 0.82 | 3.01 | 179.32 | 22.12 | 281.80 | 383.17 | 0.13 |
| | 2013 | 12.09 | 14.81 | 3.06 | 0.65 | 2.28 | 174.10 | 19.58 | 280.38 | 380.79 | 0.11 |
| | 2014 | 10.91 | 14.39 | 2.68 | 0.79 | 2.52 | 152.96 | 20.96 | 277.79 | 370.92 | 0.09 |
| | 2015 | 12.13 | 14.73 | 2.74 | 0.90 | 2.83 | 154.19 | 18.65 | 294.35 | 381.96 | 0.12 |
| | Average | 12.04 | 14.78 | 2.65 | 0.79 | 2.66 | 165.14 | 20.33 | 283.58 | 379.21 | 0.11 |

**Table 2.** The Pearson correlation coefficients between 30Min, Daily and Monthly sensible heat flux (Hs) and difference between water surface and air temperature ($\Delta T$), wind speed (U), the product of wind speed and difference between water surface and air temperature (U $\cdot$ $\Delta T$), air temperature (Ta), water surface temperature (Ts), net radiation (Rn), and rain, during pre-monsoon, monsoon and post-monsoon periods for the whole study period (2012 to 2015). The significant level of 0.01 and 0.05 are marked with ** and * respectively. The correlation coefficients between rain and Hs are estimated only at daily and monthly scales.

| variable | 30Min | | | Daily | | | Monthly | | |
|---|---|---|---|---|---|---|---|---|---|
| | pre-monsoon | monsoon | post-monsoon | pre-monsoon | monsoon | post-monsoon | pre-monsoon | monsoon | post-monsoon |
| $\Delta T$ | 0.422** | 0.564** | 0.397** | 0.613** | 0.775** | 0.548** | 0.281 | 0.922** | 0.945** |
| U | -0.144** | -0.017** | -0.003 | -0.125 | -0.248** | -0.166* | -0.162 | -0.594** | -0.511 |
| U $\cdot$ $\Delta T$ | 0.394** | 0.666** | 0.541** | 0.571** | 0.235** | 0.349** | 0.324 | 0.903** | 0.711 |
| Ta | -0.375** | -0.401** | -0.309** | -0.552** | -0.528** | -0.308** | -0.244 | -0.579** | -0.614 |
| Ts | -0.177** | -0.070** | -0.130** | -0.199** | 0.035 | -0.170* | -0.013 | 0.264 | -0.382 |
| Rn | 0.178** | -0.052** | 0.187** | -0.246** | -0.546** | -0.233** | -0.472 | -0.750** | -0.348 |
| rain | | | | 0.242** | 0.105* | 0.062 | 0.602 | 0.564** | -0.298 |

**Table 3.** The Pearson correlation coefficients between 30Min, Daily and Monthly latent heat flux (LE) and vapor pressure difference ($\triangle$e) between water surface (es) and the air (ea), wind speed (U), the product of wind speed and vapor pressure difference ($\triangle$e) between water surface and the air (U • $\Delta$e), air temperature (Ta), water surface temperature (Ts), relative humidity (Rh), net radiation (Rn), and rain, during pre-monsoon, monsoon and post-monsoon periods for the whole study period (2012 to 2015). The significant level of 0.01 and 0.05 are marked with ** and * respectively. The correlation coefficients between rain and LE are estimated only at daily and monthly scales.

| variable | 30Min pre-monsoon | monsoon | post-monsoon | Daily pre-monsoon | monsoon | post-monsoon | Monthly pre-monsoon | monsoon | post-monsoon |
|---|---|---|---|---|---|---|---|---|---|
| $\Delta$e | 0.292** | 0.215** | 0.407** | 0.214** | 0.108* | 0.288** | 0.118 | 0.024 | -0.021 |
| U | 0.502** | 0.741** | 0.723** | 0.634** | 0.798** | 0.687** | 0.791* | 0.765** | 0.568 |
| U • $\Delta$e | 0.477** | 0.649** | 0.709** | 0.544** | 0.598** | 0.682** | 0.777* | 0.400 | 0.293 |
| Ta | 0.326** | 0.374** | 0.488** | 0.239** | 0.423** | 0.633** | 0.016 | 0.627** | 0.066 |
| Ts | 0.254** | 0.301** | 0.379** | 0.097 | 0.304** | 0.341** | -0.019 | 0.419 | 0.000 |
| Rh | -0.238** | -0.202** | -0.490** | -0.310** | -0.105* | -0.421** | 0.137 | -0.042 | -0.375 |
| Rn | 0.244** | 0.243** | 0.315** | 0.111 | 0.221** | 0.301** | -0.088 | 0.503* | -0.014 |
| rain | | | | -0.052 | -0.080 | 0.088 | -0.089 | -0.352 | 0.282 |

**Table 4.** The Pearson correlation coefficients between 30Min, Daily and Monthly $CO_2$ flux and difference between water surface and air temperature ($\Delta T$), vapor pressure difference ($\triangle e$) between water surface and the air, wind speed (U), air temperature (Ta), relative humidity (Rh), water surface temperature (Ts), photosynthetic active radiation (RAR), net radiation (Rn), and rain, during pre-monsoon, monsoon and post-monsoon periods for the whole study period (2012 to 2015). The significant level of 0.01 and 0.05 are marked with ** and * respectively. The correlation coefficients between rain and $CO_2$ flux are estimated only at daily and monthly scales.

| variable | 30Min | | | Daily | | | Monthly | | |
|---|---|---|---|---|---|---|---|---|---|
| | pre-monsoon | monsoon | post-monsoon | pre-monsoon | monsoon | post-monsoon | pre-monsoon | monsoon | post-monsoon |
| $\Delta T$ | 0.035** | 0.115** | 0.078** | -0.006 | 0.252** | 0.253** | -0.598 | 0.519* | -0.386 |
| $\triangle e$ | -0.049** | -0.093** | -0.026 | 0.178** | -0.006 | 0.003 | 0.546 | -0.391 | 0.470 |
| U | -0.140** | -0.231** | -0.171** | -0.313** | -0.558** | -0.335** | -0.557 | -0.810** | 0.610 |
| Ta | -0.141** | -0.179** | -0.099** | -0.091 | -0.248** | -0.118 | 0.055 | -0.578** | 0.072 |
| Rh | 0.007 | 0.084** | 0.022 | -0.148* | 0.022 | 0.074 | -0.681 | 0.309 | -0.211 |
| Ts | -0.198** | -0.146** | -0.086** | -0.135* | -0.084 | 0.006 | -0.166 | -0.202 | -0.037 |
| PAR | -0.227** | -0.129** | -0.231** | 0.320** | 0.204** | 0.022 | 0.755* | 0.273 | -0.088 |
| Rn | -0.262** | -0.152** | -0.281** | -0.002 | -0.057 | 0.084 | 0.171 | -0.591** | 0.003 |
| rain | | | | -0.342** | 0.028 | -0.138 | -0.414 | 0.220 | 0.422 |

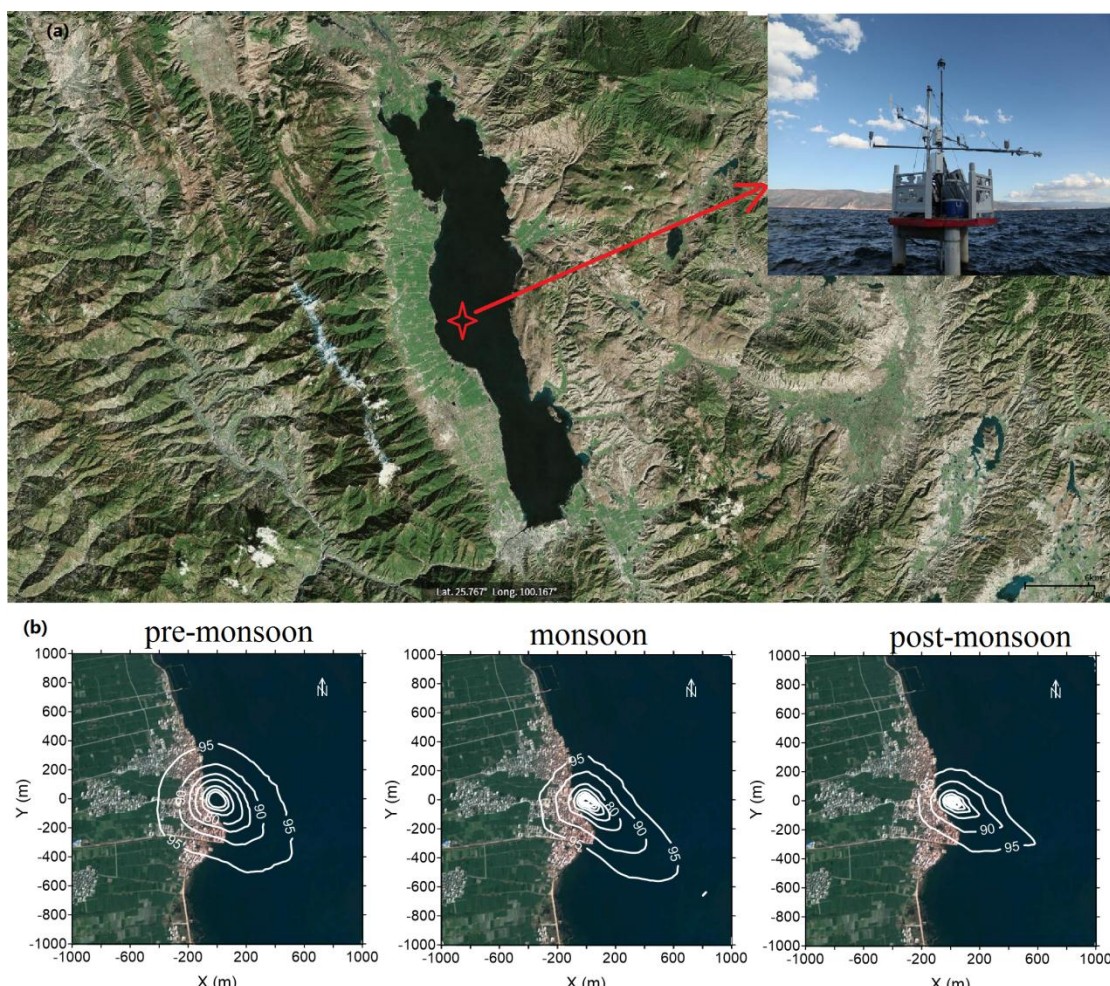

**Figure 1.** (**a**) The location of Lake Erhai (Google) and the eddy covariance measurement system (the red star denotes the flux tower); (**b**) Average footprint source over Lake Erhai flux tower during three different monsoon periods (pre-monsoon period, monsoon period and post-monsoon period) from 2012 to 2015. The maximum radius length of contour line shows the source area contributing to 95% of flux.

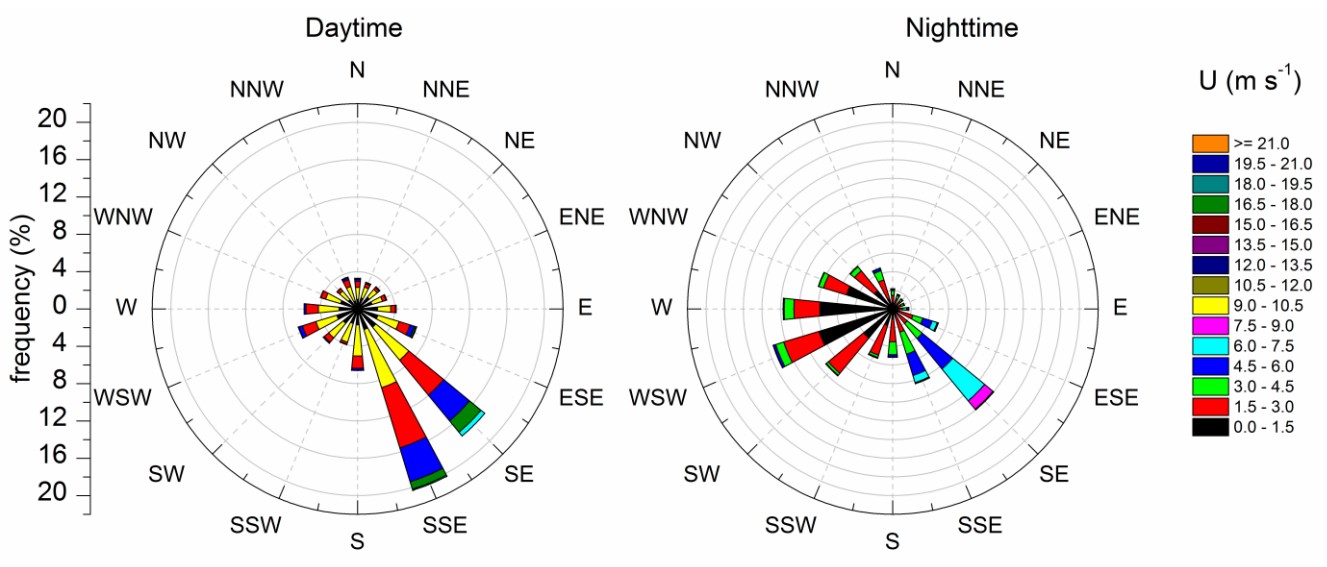

**Figure 2.** The wind rose of Erhai lake during daytime (when downward shortwave radiation larger than 20W m$^{-2}$) and nighttime (when downward shortwave radiation less than 20W m$^{-2}$) from 2012 to 2015.

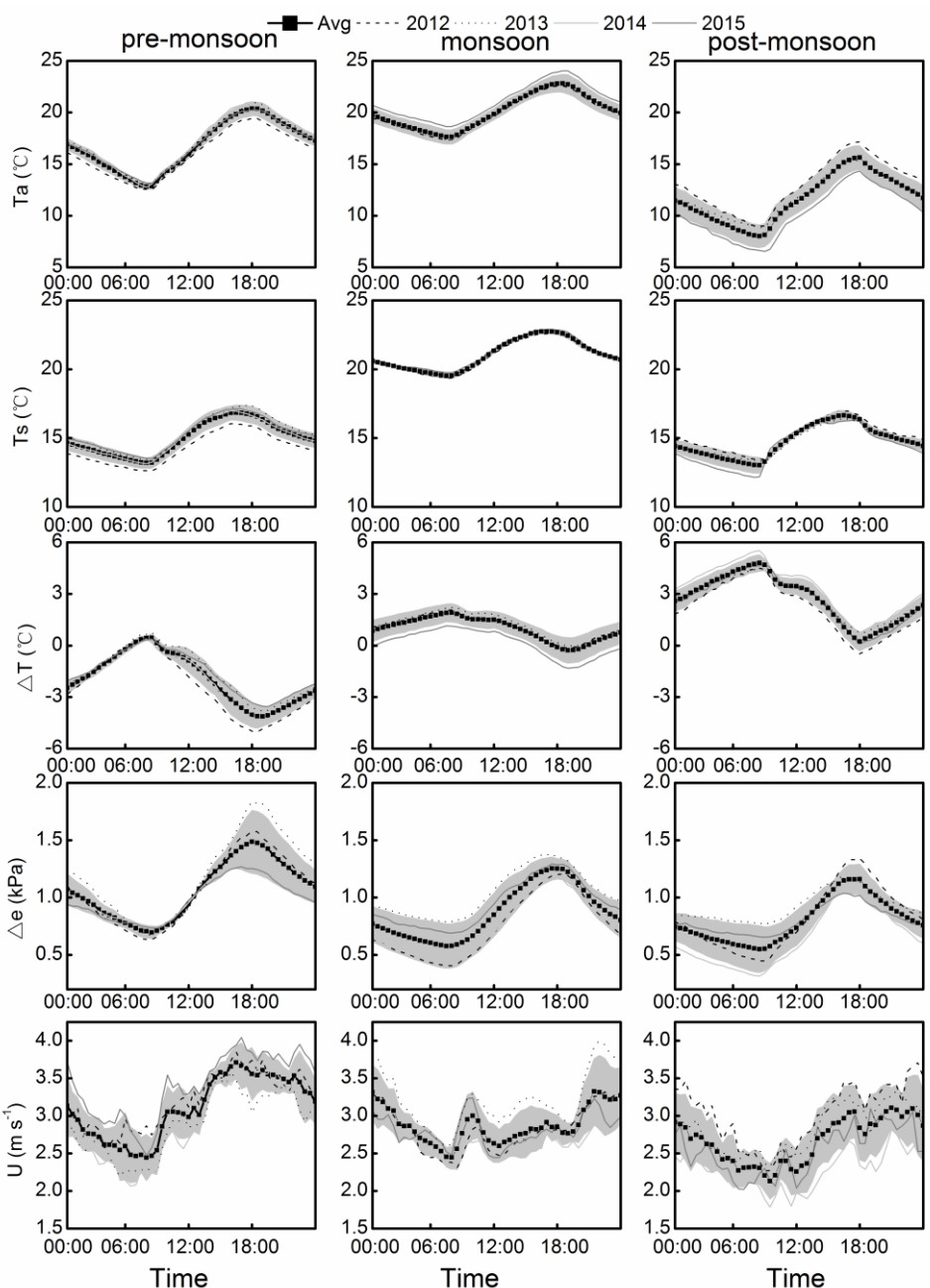

**Figure 3.** The average diurnal pattern of air temperature (Ta), water surface temperature (Ts), difference between water surface temperature and air temperature ($\Delta$T), water-air vapor pressure deficit ($\Delta$e), and wind speed (U), during pre-monsoon, monsoon and post-monsoon periods from 2012 to 2015

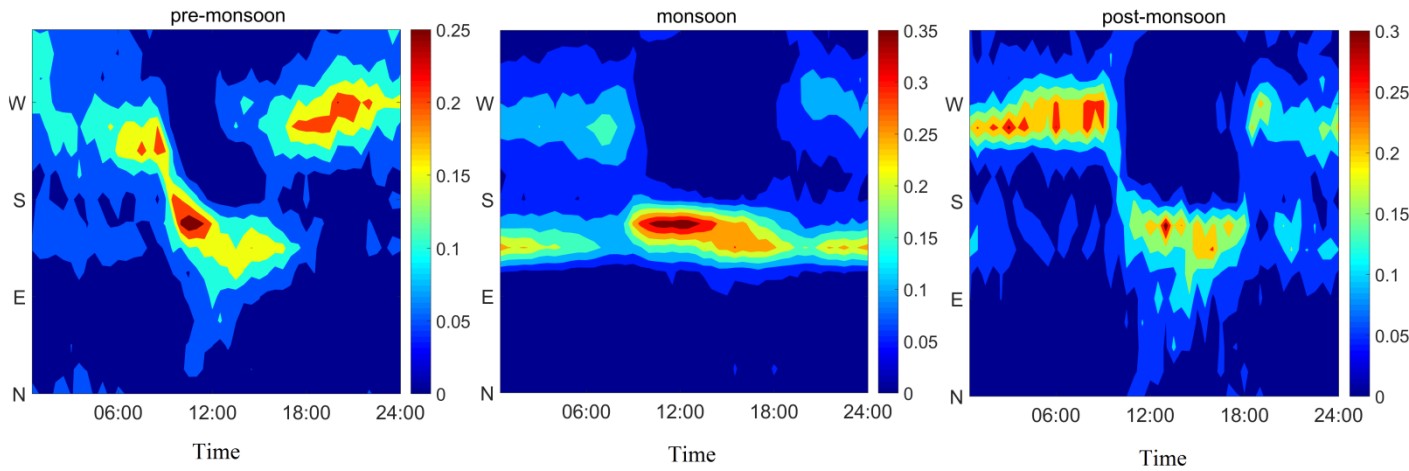

**Figure 4.** The average diurnal pattern of frequency distribution of wind direction during pre-monsoon, monsoon and post-monsoon periods from 2012 to 2015

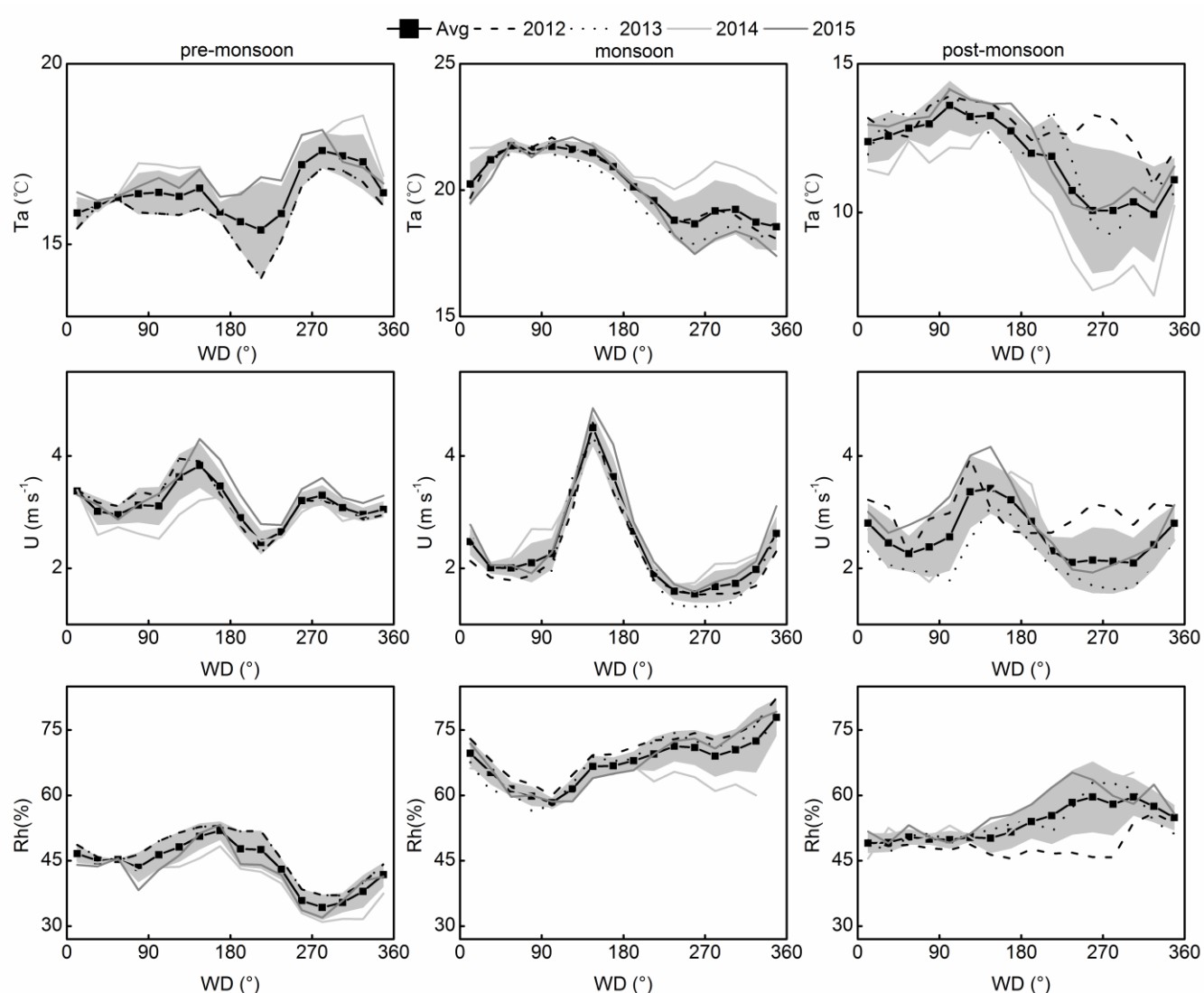

**Figure 5.** The bin-averaged wind speed (U), air temperature (Ta) and relative humidity (Rh) along with wind direction (WD) during pre-monsoon, monsoon and post-monsoon periods from 2012 to 2015. Every 22.5° was bin averaged.

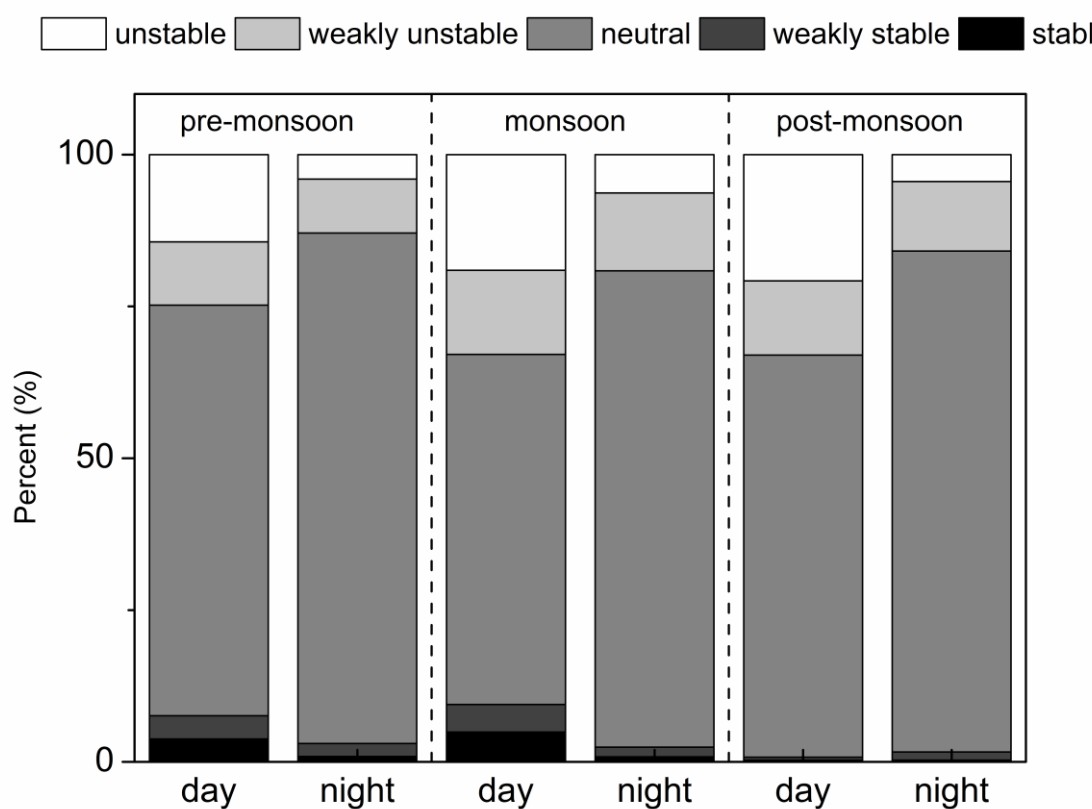

**Figure 6.** Frequency percent of stability classes in day (when the shortwave radiation $> 20$ W m$^{-2}$) and night (when the shortwave radiation $\leqslant 20$ W m$^{-2}$) during pre-monsoon, monsoon and post-monsoon periods from 2012 to 2015. The stable classes are defined as atmosphere stability ($\zeta$ =z/L, where z is the measurement height, L is the Monin-Obukhov length): stable ($\zeta > 0.1$), weakly stable ($0.05 < \zeta < 0.1$), near neutral ($-0.05 < \zeta < 0.05$), weakly unstable ($-0.1 < \zeta < 0.05$) and unstable ($\zeta < -0.1$).

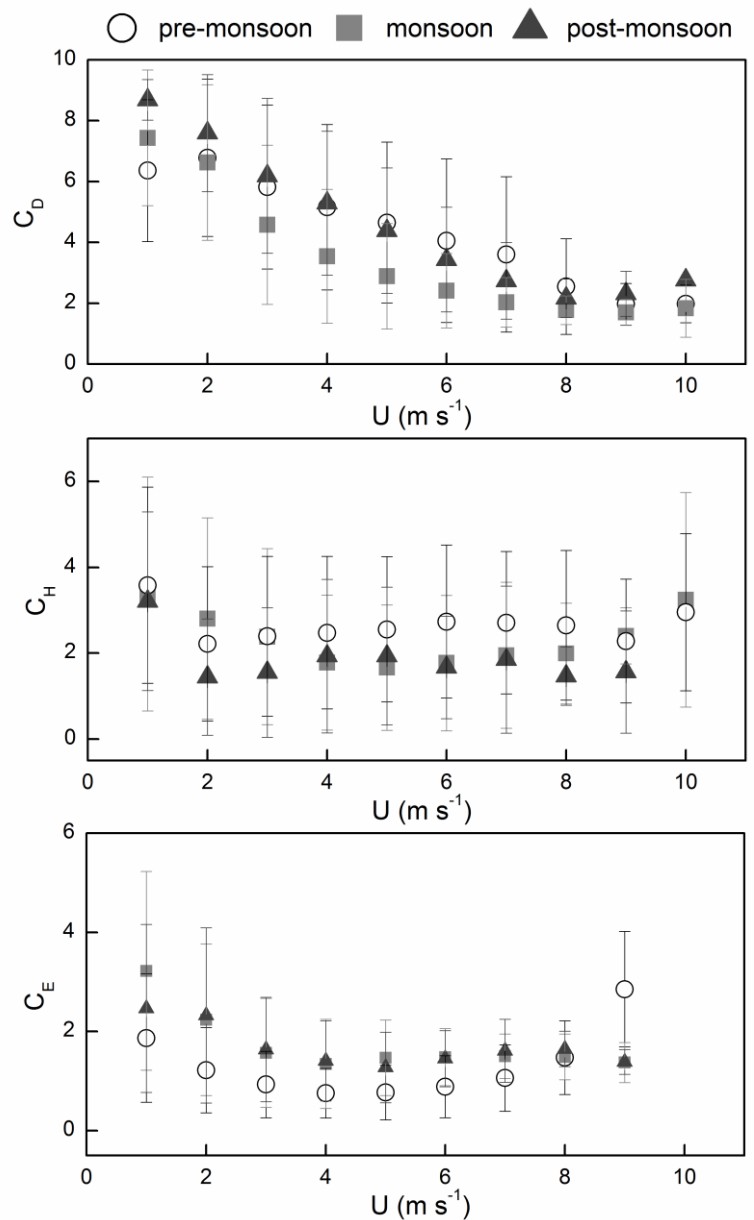

**Figure 7.** The relationship between bin (bin width 1 m/s) averaged drag coefficient (the momentum bulk transfer coefficient, $C_D$), Dalton number (the heat bulk transfer coefficient, $C_H$) and Stanton number (the moisture bulk transfer coefficient, $C_E$) and wind speed (U) during pre-monsoon, monsoon and post-monsoon periods for whole study period. The error bars show the $\pm 1$ standard deviation of the average value.

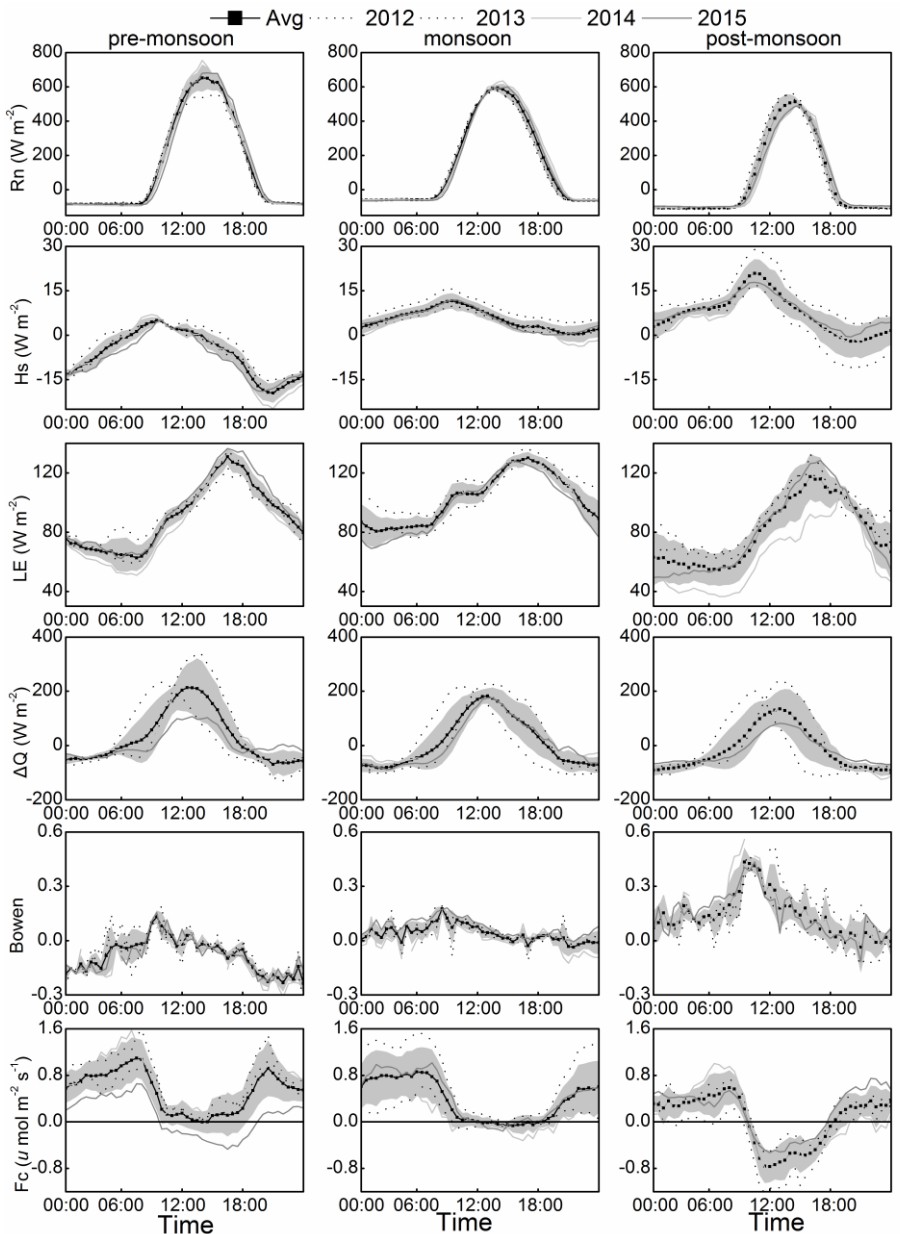

**Figure 8.** The average diurnal patterns of energy balance components (the net radiation flux, Rn; the sensible heat flux, Hs; the latent heat flux, LE; and the storage heat flux in the lake, $\triangle$ Q) , Bowen ratio (Bowen) and $CO_2$ flux (Fc) during pre-monsoon, monsoon and post-monsoon periods from 2012 to 2015.

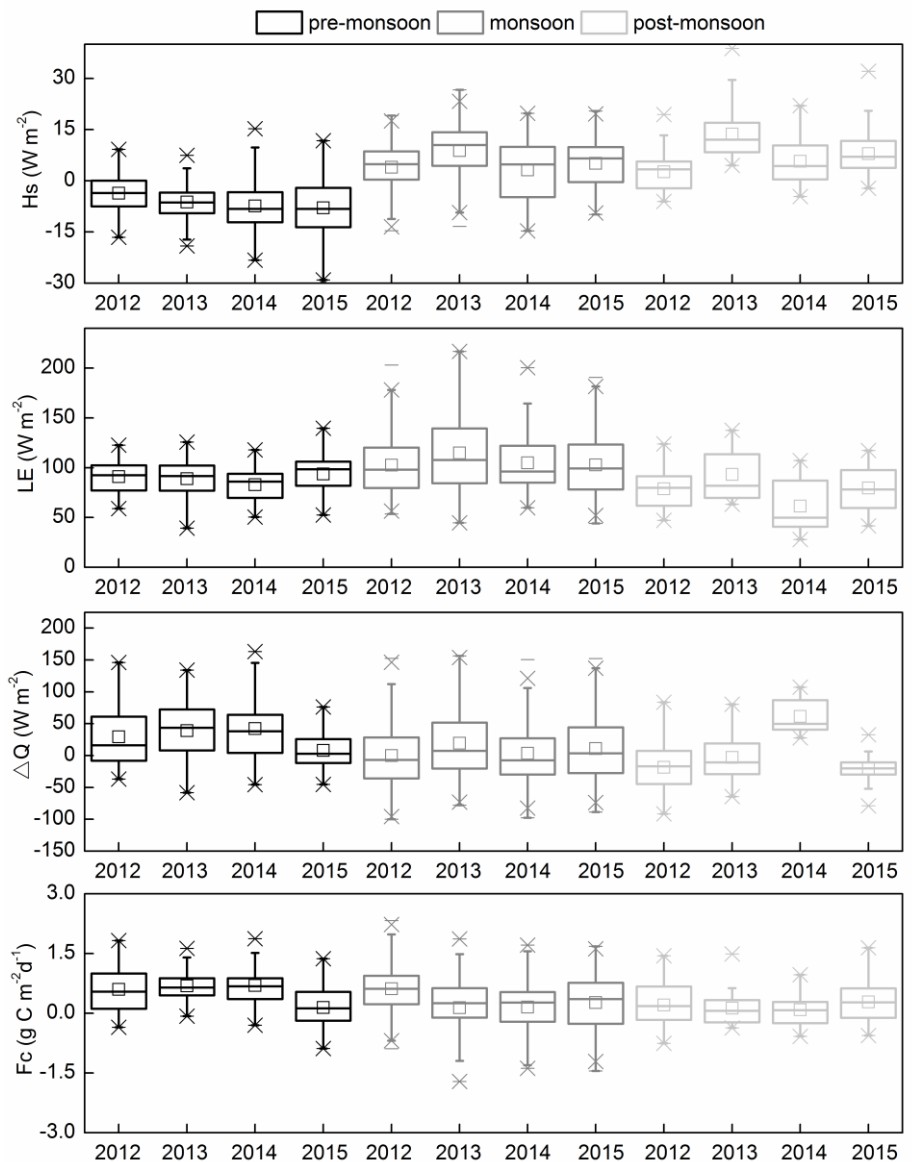

**Figure 9.** The boxplot of daily average sensible heat flux (Hs), latent heat flux (LE), the storage heat flux in the lake ($\triangle$Q) and $CO_2$ flux (Fc) during pre-monsoon, monsoon and post-monsoon periods from 2012 to 2015. The upper and lower limits of the box represent the 75% and 25% percentiles; The horizontal line in each box represent the 1.5 Inter-Quartile Range of the upper and lower quartile; The band inside the box is the median; The squares inside the box represent the average value; The cross-hatches represent the 1% and 99% percentiles; The ends of whiskers represent the maximum and minimum value.

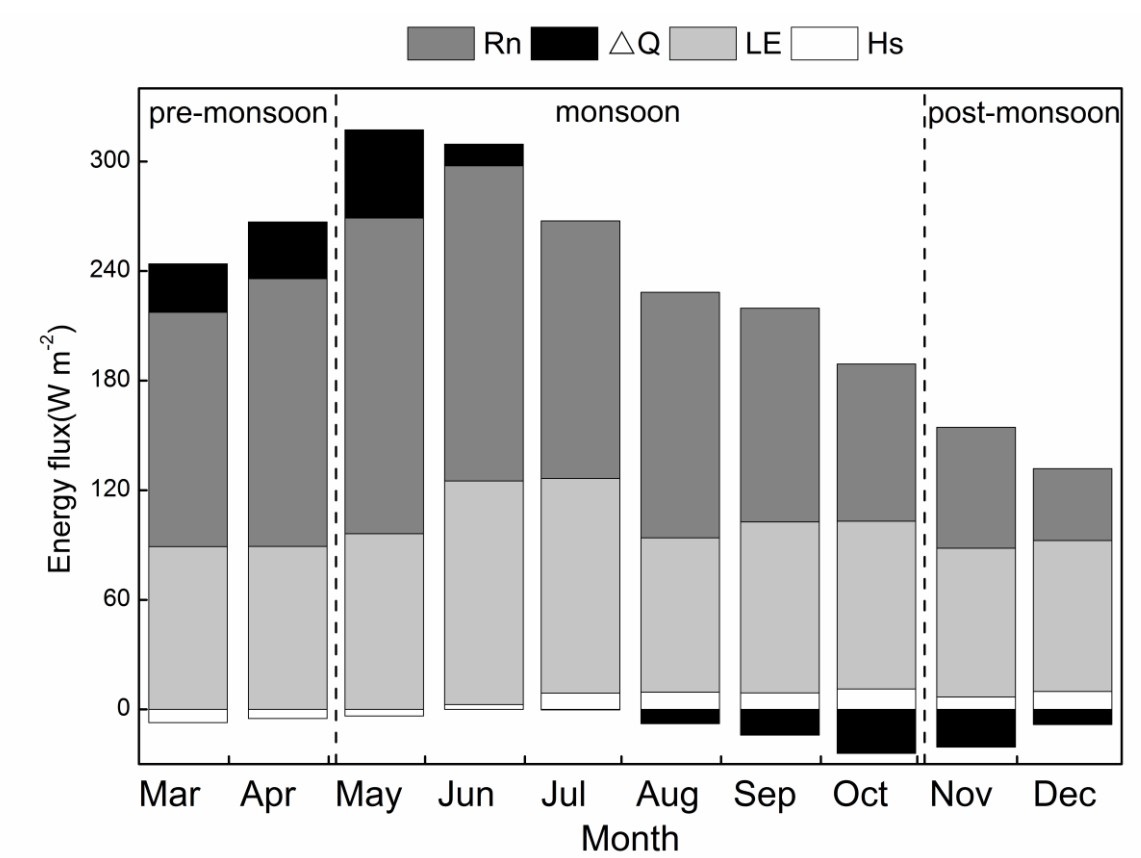

**Figure 10.** The monthly average energy fluxes (the storage heat flux in the lake, $\triangle Q$; the net radiation flux, Rn; the latent heat flux, LE; and the sensible heat flux, Hs) during pre-monsoon, monsoon and post-monsoon periods from 2012 to 2015.