# Peer review of "The monsoon effect on energy and carbon exchange processes over a highland lake"

_Atmospheric Chemistry and Physics, 2018_

## Referee Comment (RC1) · Anonymous Referee #1 · 23 Mar 2018

**Review of the manuscript by Du et al**

**Title: The monsoon effect on energy and carbon exchange processes over a highland lake in southwest of China.**

The study presents several years of energy and carbon dioxide fluxes measured at Lake Erhai. The focus is to investigate the different dynamics of fluxes and their drivers in three sub-periods, e.g. pre-monsoon, monsoon and post-monsoon. The dataset is interesting, the framework analysis and results/discussion comprehensive and well written. I can recommend the final publication in ACP after the following comments are properly addressed:

1) I suggest to re-structured the Results chapter. In my opinion subchapters 3.1, 3.2 and 3.3 could be merged and shortened in one sub-chapter related to environmental/atmospheric conditions. So many details on diurnal and seasonal variation of each radiation components is not so interesting and they could also be omitted. Instead, the focus could be more on net radiation, heat storage and turbulent fluxes (H and LE).

2) The $CO_2$ (and LE) flux data are measured with an open-path (OP) sensor. Generally, I think the use of OP should be avoid over ecosystem where fluxes are expected to be quite small. The authors should add some discussion on this point, and also try to acknowledge the uncertainty due to WPL correction and the potential sign change in $CO_2$ fluxes.

3) Related to the previous point, Have the authors made some independent measurements supporting the net uptake of $CO_2$ for certain periods. Did the authors measure, for example, $pCO_2$ in the water?

Minor comments:

- P1L24. "…is the main driver for Hs…" and not "drivers".
- P2L20. Replace "influence" with "affects".
- P5L10. "Webb-Pearman-Leuning".
- P5L9. Please explain what is the circular correlation procedure" or give the reference.
- Eq. 1. Use $\Delta t$ to indicate the time difference.
- P6.L26. I guess 600 meters and 400 meters.
- P6L1. "….post-monsoon period…"
- P6L2. I would rephrase as "…filtered based on the footprint analysis."
- P6L14-15. The sentence "The diurnal……period." is not clear, please rephrase it.

- P6L34. Monsoon is sometime written with capital letters and sometimes not. Please write it consistently trough the text.
- P7L27). Ta was already defined above.
- P8L4-5. I would rephrase it as "The atmospheric surface layer is mainly near neutral stratified during the three study periods.."
- Sect 3.3. My suggestion is to remove this section.
- P10L15-20. Would the authors expect that there is no phytoplankton during the pre-monsoon period? What is the uncertainty associated with the $CO_2$ fluxes? Could the authors show the error bars or the confidence intervals in Figure 7? Are the around midday $CO_2$ fluxes significantly different than zero?
- P11L5. Please correct "period".
- P11L30. The sentence is not clear. From where do the authors see a correlation between $R_n$ and $U\Delta T$?
- P12L29-34. The text in this paragraph is somehow a repetition of what has been said above. Please remove/merge the text.
- P13L21-26. The rain is also enhancing the transport of carbon from land/catchment areas to the water system (lateral fluxes) enhancing DOC in the water (see for example Pumpanen et al., 2014) and potentially the $pCO_2$.

References:

Pumpanen, J., Linden, A., Miettinen, H., Kolari, P., Ilvesniemi, H., Mammarella, I., Hari, P., Nikinmaa, E., Heinonsalo, J., Back, J., Ojala, A., Berninger, F., and Vesala, T., 2014. Precipitation and net ecosystem exchange are the most important drivers of DOC flux in upland boreal catchments, J. Geophys. Res. Biogeosci., 119, 1861–1878, doi:10.1002/2014JG002705

---

## Referee Comment (RC2) · Anonymous Referee #2 · 1 Jun 2018

General comments

The paper presents interesting measurements of lake energy and CO2 fluxes from a high altitude monsoon site. The site location adds novelty to the study and further advances our understanding of regional differences and similarities in terms of lake-air interaction.

The comparison of temporal patterns during, after and before monsoon is novel and interesting to see. The reasoning and explanation of the behavior is physically sound

However, my main concerns is the analysis regarding the fluxes. In my opinion the correlation analysis does not add anything to our physical understanding of the forcing

mechanisms for the energy and CO2 fluxes. It is for instance well established that both U and the gradient controls H and E. This does not need further investigation. For CO2 the situation is slightly more complicated, but to say anything meaningful about the flux variation the CO2 concentration of the water would have to be measured. The gradient is thermodynamic forcing and the efficiency of the exchange is determined by the transfer velocity.

In term of H, the variation in U is very small so there is no surprise that the variation in sensible heat flux is due to variations in the temperature gradient. E is determined by both dq and U, here dq and U seem to be well correlated (you can calculate the correlation coefficient) which then explains your results.

My suggestion is, instead of the correlation analysis, compute the Stanton and Dalton numbers for the different cases, and analyze these results. You also need to add more references on the CO2 lake fluxes in the introduction/background and discuss your results in terms of these previous findings. Also, add background on e.g. transfer velocity, surface renewal etc. to at least conceptually put your results in this context.

In general, I also think that the figures showing results for all years should be combined into one single plot as they mostly are very similar. Are there any statistical significant difference between the years? Some measure of variation would also be interesting to include.

Specific comments

Page 4, Line 4: water level should be stated as water depth, not height above sea level as is stated now (?)

P4, L 10, how large is the annual precipitation, and in average how much during the different periods (pre, during, after monsoon)?

P4 L15: How are the booms oriented on which you placed the eddy covariance sensors? Did you consider any flow distortion effects from the platform (it is a quite solid

construction)?

P4, L 22: CS616 cannot be correct, this is a water content reflectometer

P4 L24 : Ts seem to be defines as skin temperature. How does this compare with the standard bulk water temperature at 0.5 m depth?

P 4, L26: Was the entire field of view located over water for the net radiation sensor? You can estimate this with the measurement height and the length of the boom. It appears from the photo in Fig 1 that the sensor might include also some parts of the platform which would affect your Ts values.

P4 L31, specify that it is cup anemometer.

P5 L 4: what do you mean by "also filtered"? AGC 40 is quite low limit, why not set a higher limit?

P5, L6, what is the averaging time?

P5, L8: if you are using block average you are not detrending. What do you mean by "circular correlation procedure"?

P5 L11: Add reference for the high frequency correction method. Was this a severe problem?

P5 L24: It would be useful with a wind rose to illustrate the prevailing wind direction.

P6 L4: how many data is included in the final data set? How are the data distributed over the different years? Is there any season that has significantly less data than the others?

P6 Section 3: This text is very compact, please separate into paragraphs.

P7, L6: Do you have any more supporting evidence that it is a sea/land breeze circulation? Any land site in the vincinity? Is there any previous studies on this for this site? A single surface observation is in my opinion not enough to support this, it is indicative

but not complete shown. As you state on line 11 that strong synoptic (?) west wind during night, this might be explanation for the night wind direction and not land breeze.

P9 L1, "resulted in a lower abledo" compared to what?

P9 L8 The Katsaros reference in the reference list is wrong, please use the correct one.

Sections 3.4 to 3.8 please see the general comments

Figure 8: No data for monsoon and post monsoon for 2014?

---

## Author Comment (AC1) · 29 Jun 2018

Many Thanks for reviewer's valuable comments and suggestions, which help a lot to improve our manuscript. All the revisions have been marked with red color in the manuscript. The responses point by point are as following:

The study presents several years of energy and carbon dioxide fluxes measured at Lake Erhai. The focus is to investigate the different dynamics of fluxes and their drivers in three sub-periods, e.g. pre-monsoon, monsoon and post-monsoon. The dataset is interesting, the framework analysis and results/discussion comprehensive and well written. I can recommend the final publication in ACP after the following comments

are properly addressed: 1) I suggest to re-structured the Results chapter. In my opinion subchapters 3.1, 3.2 and 3.3 could be merged and shortened in one sub-chapter related to environmental/atmospheric conditions. So many details on diurnal and seasonal variation of each radiation components is not so interesting and they could also be omitted. Instead, the focus could be more on net radiation, heat storage and turbulent fluxes (H and LE).

Answer: We agree with your point. Details about diurnal patters of meteorological variables are too many to be interesting for readers. Subchapter 3.3 about the diurnal variation of radiation components has been removed. The results about the atmospheric conditions have been shortened.

2) The $CO_2$ (and LE) flux data are measured with an open-path (OP) sensor. Generally, I think the use of OP should be avoid over ecosystem where fluxes are expected to be quite small. The authors should add some discussion on this point, and also try to acknowledge the uncertainty due to WPL correction and the potential sign change in $CO_2$ fluxes.

Answer: In the open-path analyzers, temperature and pressure vary with ambient conditions, so the Webb-Pearman-Leuning (WPL) density corrections (Webb et al. 1980) is necessary to correct for the fluctuations. However, through the standard data processing and quality control, the data measured by open and closed path systems are in quite good agreement. So the open path system is widely used to study the turbulent exchange process between the lake surface and the atmosphere, e.g., an open-path EC system containing LI-7500A was installed to measure LE, Hs, and $CO_2$ flux in Western Lake Erie (Shao et al., 2015), the turbulent exchange process is studied over a small lake in the Nam Co basin on the Tibetan Plateau based on the measurement with an open-path infrared gas analyzer (LI 7550, LI-COR,Inc.) (Wang et al., 2017), Goldbach and Kuttler (2015) also measured the turbulent fluxes over a suburban reservoir in Germany with an infrared open-path analyzer (LI-7500, LI-COR,Inc.). The WPL correction has a large effect on $CO_2$ fluxes because it could cause sign change in

$CO_2$ fluxes. We evaluate the uncertainty of WPL correction on $CO_2$ flux based on the raw data from October of 2015. The daily average $CO_2$ flux with and without WPL correction is 0.91±1.95 g C m-2 d-1 and -0.25±2.69 g C m-2 d-1, respectively. This information has been supplemented in the manuscript. References: Goldbach, A., and Kuttler, W.: Turbulent Heat Fluxes above a Suburban Reservoir: A Case Study from Germany, J. Hydrometeorol., 16, 244-260, doi: 10.1175/JHM-D-13-0159.1, 2015. Shao, C., Chen, J., Stepien, C. A., Chu, H., Ouyang, Z., Bridgeman, T. B., Czajkowski, K. P., Becker, R. H., and John, R.: Diurnal to annual changes in latent, sensible heat, and $CO_2$ fluxes over a Laurentian Great Lake: A case study in Western Lake Erie, J. Geophys. Res-Biogeo., 120, 1587-1604, doi: 10.1002/2015JG003025, 2015. Wang, B., Ma, Y., Ma, W., and Su, Z., Physical controls on half-hourly, daily, and monthly turbulent flux and energy budget over a high-altitude small lake on the Tibetan Plateau, J. Geophys. Res. Atmos., 122, 2289–2303, doi:10.1002/2016JD026109, 2017.

3) Related to the previous point, Have the authors made some independent measurements supporting the net uptake of $CO_2$ for certain periods. Did the authors measure, for example, $pCO_2$ in the water?

Answer: Sorry, the measurements on $pCO_2$ are still absent now. However, some researchers have conducted biochemical measurements in Lake Erhai, and the seasonal variation of of Chl a and phytoplankton concentration has been reported (Yu et al., 2014). This could be seen as a support for the $CO_2$ uptake at some periods. This information has been added in the manuscript. In future, measurements of $CO_2$ partial pressure will be supplemented to improve understanding on $CO_2$ exchange rate over Lake Erhai. Yu, G., Jiang, Y., Song, G., Tan, W., Zhu, M., and Li, R.: Variation of Microcystis and microcystins coupling nitrogen and phosphorus nutrients in Lake Erhai, a drinking-water source in Southwest Plateau, China, Environ. Sci Poll. R. Int., 21, 9887-9898, 2014.

Minor comments: - P1L24. "...is the main driver for Hs..." and not "drivers".

Answer: It has been corrected.

- P2L20. Replace "influence" with "affects".

Answer: It has been replaced.

- P5L10. "Webb-Pearman-Leuning".

Answer: The "webb" has been capitalized as "Webb".

- P5L9. Please explain what is the circular correlation procedure" or give the reference.

Answer: The circular correlation procedure is one of the methods EddyPro provided to compensate the time lags between anemometric variables and gas analyzer measurements, which determines the time lag that maximizes the covariance of two variables, within a window of plausible time lags (Fan et al., 1990). This has been supplemented in the manuscript.

- Eq. 1. Use Dt to indicate the time difference.

Answer: The equation has been corrected.

- P5.L26. I guess 600 meters and 400 meters.

Answer: Sorry for the missing of the units, which have been supplemented.

- P6L1. "….post-monsoon period…"

Answer: The"Period"has been revised as "period".

- P6L2. I would rephrase as "…filtered based on the footprint analysis."

Answer: Thanks, the sentence has been revised according to the suggestion.

- P6L14-15. The sentence "The diurnal……period." is not clear, please rephrase it.

Answer: This sentence has been revised as "The diurnal mean Ta is the largest during monsoon period, second during pre-monsoon period and smallest during postmonsoon period".

- P6L34. Monsoon is sometime written with capital letters and sometimes not. Please write it consistently trough the text.

Answer: This word has been written uniformly as "monsoon" throughout the manuscript.

- P7L27). Ta was already defined above. Answer: Thanks for your remind. The definition has been removed from here.

- P8L4-5. I would rephrase it as "The atmospheric surface layer is mainly near neutral stratified during the three study periods.."

Answer: Thanks for your suggestion. The sentence has been revised accordingly.

- Sect 3.3. My suggestion is to remove this section.

Answer: We accept this suggestion and remove section 3.3. We agree that the seasonal variation of radiation components is not so interesting as it's mainly caused by solar elevation angle. Our study also shows slight difference between pre-monsoon period and post-monsoon period, so it's not much meaningful.

- P10L15-20. Would the authors expect that there is no phytoplankton during the pre-monsoon period? What is the uncertainty associated with the $CO_2$ fluxes? Could the authors show the error bars or the confidence intervals in Figure 7? Are the around midday $CO_2$ fluxes significantly different than zero?

Answer: The seasonal fluctuation of phytoplankton in Lake Erhai has been reported by some researchers. Yu et al. (2014) observed that the concentration of Chl a and phytoplankton in Lake Erhai were higher in mid-summer and autumn and fell down from winter until April. This reference has been added in the manuscript. The error bars have been added in Figure 7, which could show the uncertainty range for $CO_2$ fluxes. The significant carbon uptake could be observed at midday time, but the peak

rate varied from year to year. The peak diurnal average CO2 uptake ranged from 0.05±0.73 umol m-2 s-1 to 0.53±1.66 umol m-2 s-1 during monsoon period, and from 0.74 ±0.89 umol m-2 s-1 to 1.62±1.52 umol m-2 s-1 during post-monsoon period from 2012 to 2015. This has been added in the manuscript too.

- P11L5. Please correct "period".

Answer: The mistake has been corrected.

- P11L30. The sentence is not clear. From where do the authors see a correlation between Rn and UDT?

Answer: Sorry. The "Rn" in the sentence should be "Hs". It has been corrected in the manuscript.

- P12L29-34. The text in this paragraph is somehow a repetition of what has been said above. Please remove/merge the text.

Answer: The text has been merged into the last paragraph.

- P13L21-26. The rain is also enhancing the transport of carbon from land/catchment areas to the water system (lateral fluxes) enhancing DOC in the water (see for example Pumpanen et al., 2014) and potentially the pCO2. References: Pumpanen, J., Linden, A., Miettinen, H., Kolari, P., Ilvesniemi, H., Mammarella, I., Hari, P., Nikinmaa, E., Heinonsalo, J., Back, J., Ojala, A., Berninger, F., and Vesala, T., 2014. Precipitation and net ecosystem exchange are the most important drivers of DOC flux in upland boreal catchments, J. Geophys. Res. Biogeosci., 119, 1861– 1878, doi:10.1002/2014JG002705

Answer: Thanks for sharing the opinion with us. The rain not only could promote CO2 uptake by enhancing the nutrients but also promote CO2 emission by enhancing pCO2 in the water, which could explain both the negative and positive correlation coefficients between rain and CO2 flux during different periods in our study. This has been added in our manuscript.

Please also note the supplement to this comment:
https://www.atmos-chem-phys-discuss.net/acp-2018-14/acp-2018-14-AC1-supplement.pdf
* * *
[Figure]

**Supplement:**

[revised manuscript text omitted]

---

## Author Comment (AC2) · 29 Jun 2018

Many Thanks for reviewer's valuable comments and suggestions, which help a lot to improve our manuscript. All the revisions have been marked with red color in the manuscript. The responses point by point are as following:

General comments The paper presents interesting measurements of lake energy and $CO_2$ fluxes from a high altitude monsoon site. The site location adds novelty to the study and further advances our understanding of regional differences and similarities in terms of lake-air interaction. The comparison of temporal patterns during, after and before monsoon is novel and interesting to see. The reasoning and explanation of the

behavior is physically sound However, my main concerns is the analysis regarding the fluxes. In my opinion the correlation analysis does not add anything to our physical understanding of the forcing mechanisms for the energy and CO2 fluxes. It is for instance well established that both U and the gradient controls H and E. This does not need further investigation. For CO2 the situation is slightly more complicated, but to say anything meaningful about the flux variation the CO2 concentration of the water would have to be measured. The gradient is thermodynamic forcing and the efficiency of the exchange is determined by the transfer velocity. In term of H, the variation in U is very small so there is no surprise that the variation in sensible heat flux is due to variations in the temperature gradient. E is determined by both dq and U, here dq and U seem to be well correlated (you can calculate the correlation coefficient) which then explains your results. My suggestion is, instead of the correlation analysis, compute the Stanton and Dalton numbers for the different cases, and analyze these results. You also need to add more references on the CO2 lake fluxes in the introduction/background and discuss your results in terms of these previous findings. Also, add background on e.g. transfer velocity, surface renewal etc. to at least conceptually put your results in this context. In general, I also think that the figures showing results for all years should be combined into one single plot as they mostly are very similar. Are there any statistical significant difference between the years? Some measure of variation would also be interesting to include.

Answer: Thanks for your valuable comments and suggestions. Although it is well understood that latent heat fluxes and sensible heat fluxes are primarily controlled by wind speed and water vapor or temperature gradients, the factors controlling turbulent fluxes vary among lakes and lake-air interaction could be affected by the lake characteristics, including lake sizes, lake depths, lake dimensions, as well as geographic location (Liu et al., 2009). A weak correlation between U and latent heat flux was found over Ross Barnett Reservoir (Liu et al., 2012) and a small lake in Finland (Nordbo et al., 2011) but a positive correlation was observed for the Great Slave Lake for wind speeds larger than a particular threshold value (Blanken et al., 2003). Water vapor gradient was found

to play a dominant role in determining energy fluxes under conditions involving large water vapor gradients, while atmospheric stability becomes significant under small water vapor gradients (Zhang and Liu, 2014). A close correlation was found between net radiation and evaporation (Yao, 2009) but a weak correlation was also found in some lakes (Liu et al., 2012). Lakes with different characteristics may respond differently to various physical forcing. Therefore, more deep study is needed to improve understanding of process controlling lake-air turbulent fluxes. For CO2 fluxes, EC technique could provide long-term continuous measurement compared to traditional methods, e.g., floating chamber and boundary layer techniques. EC measurements have been used also to develop new empirical models of gas transfer velocity k using simultaneous measurements of CO2 partial pressure at the water surface (Mammarella et al., 2014). However, due to the lack of measurements of CO2 partial pressure in our present observation, we didn't study the gas exchange rate. In future, measurements of CO2 partial pressure will be supplemented to develop further study on the variation of gas exchange rate and its controlling meteorological variables. We don't think the poor correlation between U and Hs is attributed to the variation of U is small. Although U is found to have a weak effect on Hs, a strong effect of U on LE is observed. We have calculated the correlation coefficient between U and deltae and the results show large variation, which is not in agreement with the correlation coefficient between U and LE. Therefore, so we don't think the close correlation between U and LE could be explained by the correlation between U and deltae. Some studies have reported the large effect of U on LE especially in some small lakes (Assouline et al., 2008; Wang et al., 2017). This has been added in the manuscript. The correlation between U and âŰşe, and LE are shown in a table, which has been uploaded.

Correlation analysis is a widely used statistical method. Plenty of scientists have applied this method to investigate the relationship between lake-atmosphere turbulent fluxes and meteorological variables (Nordbo et al., 2011; Liu et al., 2012; Zhang and Liu, 2014; Goldbach and Kuttler, 2015). In order to compare with other studies on lake and atmosphere interaction, we retain the results analysis with the use of this method.

[Figure]

Besides, the Stanton and Dalton numbers have been estimated and the analyzed in the manuscript (Figure 8). More references on lake $CO_2$ fluxes have been added in the introduction and compared with our results in discussion. Meantime, the introduction on transfer velocity has been added in background too. All figures have been changed by combining four years into a single plot. The average and the standard error of meteorological variables and fluxes for four years period have been calculated and shown in individual figures. A boxplot is also added to show the variation of turbulent fluxes in different years (Figure 9), and the results have been analyzed in the manuscript.

References: Assouline, S., S. W. Tyler, J. Tanny, S. Cohen, E. Bou-Zeid, M. B. Parlange, and G. G. Katul (2008), Evaporation from three water bodies of different sizes and climates: Measurements and scaling analysis, Adv. Water Resour., 31(1), 160–172, doi:10.1016/j.advwatres.2007.07.003.

Blanken, P. D., W. R. Rouse, and W. M. Schertzer (2003), Enhancement of evaporation from a large northern lake by the entrainment of warm, dry air, J. Hydrometeorol., 4(4), 680–693, doi:10.1175/1525-7541(2003)004<0680:eoefal>2.0.co;2.

Goldbach, A., and Kuttler, W.: Turbulent Heat Fluxes above a Suburban Reservoir: A Case Study from Germany, J. Hydrometeorol., 16, 244-260, doi: 10.1175/JHM-D-13-0159.1, 2015.

Liu, H., Y. Zhang, S. Liu, H. Jiang, L. Sheng, and Q. L. Williams (2009), Eddy covariance measurements of surface energy budget and evaporation in a cool season over southern open water in Mississippi, J. Geophys. Res., 114, D04110, doi:10.1029/2008JD010891.

Liu, H., Q. Zhang, and G. Dowler (2012), Environmental controls on the surface energy budget over a large southern inland water in the United States: An analysis of one-year eddy covariance flux data, J. Hydrometeorol., 13(6), 1893–1910, doi:10.1175/JHM-D-12-020.1.

Mammarella, I., et al. (2015), Carbon dioxide and energy fluxes over a small boreal lake in Southern Finland, J. Geophys. Res. Biogeosci., 120, 1296–1314, doi:10.1002/2014JG002873.

Nordbo, A., S. Launiainen, I. Mammarella, M. Leppäranta, J. Huotari, A. Ojala, and T. Vesala (2011), Long-term energy flux measurements and energy balance over a small boreal lake using eddy covariance technique, J. Geophys. Res., 116, D02119, doi:10.1029/2010JD014542.

Wang, B., Y. Ma, W. Ma, and Z. Su (2017), Physical controls on half-hourly, daily, and monthly turbulent flux and energy budget over a high-altitude small lake on the Tibetan Plateau, J. Geophys. Res. Atmos., 122, 2289–2303, doi:10.1002/2016JD026109.

Yao, H. X. (2009), Long-term study of lake evaporation and evaluation of seven estimation methods: Results from Dickie Lake, South-Central Ontario, Canada, J. Water Resour. Prot., 1(2), 59–77

Zhang, Q., and H. Liu (2014), Seasonal changes in physical processes controlling evaporation over inland water, J. Geophys. Res. Atmos., 119, 9779–9792, doi:10.1002/2014JD021797.

Specific comments Page 4, Line 4: water level should be stated as water depth, not height above sea level as is stated now (?)

Answer: The water depth of Lake Erhai has been introduced in the last paragraph "The water depth of the lake varies from 10 and 20.7 m". "Water level" here indeed means the height above sea level.

P4, L 10, how large is the annual precipitation, and in average how much during the different periods (pre, during, after monsoon)?

Answer: The average precipitation of the whole year and three different periods has been added in the manuscript.

P4 L15: How are the booms oriented on which you placed the eddy covariance sensors? Did you consider any flow distortion effects from the platform (it is a quite solid construction)?

Answer: The EC sensors are mounted on a pipe orienting to the prevailing wind direction (southeast) at the height of 2 m above the platform. This has been added in the manuscript. The platform is about 1 m above the water surface with a radius of 1 m. The platform is supported by three piers distributing in triangle shape in order to make the air flow pass conveniently (Figure 1). As the air flow could pass the platform easily due to the structure of the platform, the effect of flow distortion is minor and could be neglected. These measurement details could also be referred to our previous studies (Liu et al., 2015).

P4, L 22: CS616 cannot be correct, this is a water content reflectometer

Answer: The temperature probes should be "model 109-L (Campbell Scientific, Inc)". The mistake has been corrected.

P4 L24 : Ts seem to be defines as skin temperature. How does this compare with the standard bulk water temperature at 0.5 m depth?

Answer: We compared Ts and the water temperature at 0.5 m (T0.5) depth for the year of 2012. The result shows Ts and T0.5 have similar temporal variation and T0.5 is slightly higher than Ts. the average of 30-min Ts and T0.5 in 2012 are 17.2±4.5°C and 17.6±4.6°C respectively.

P 4, L26: Was the entire field of view located over water for the net radiation sensor? You can estimate this with the measurement height and the length of the boom. It appears from the photo in Fig 1 that the sensor might include also some parts of the platform which would affect your Ts values.

Answer: A figure is uploaded in the attachment which could show the position of the radiation sensors more clearly. The sensors are marked with red circle in the figure.

The radiation sensors are entirely over the water surface. Figure The observation system over Erhai Lake

P4 L31, specify that it is cup anemometer.

Answer: Thanks for your suggestion. This information has been supplemented.

P5 L 4: what do you mean by "also filtered"? AGC 40 is quite low limit, why not set a higher limit?

Answer: It means not only the data points outside the normal range but also those flagged with AGC larger than 40 are filtered. The threshold of AGC is determined by analyzing the relationship between AGC and precipitation. The results show the AGC is usually more than 40 when there is precipitation, indicating the lower data quality in this case.

P5, L6, what is the averaging time?

Answer: The averaging time is 30 minutes, which has been added in the manuscript.

P5, L8: if you are using block average you are not detrending. What do you mean by "circular correlation procedure"?

Answer: Sorry for the mistake about detrending, it has been corrected . The circular correlation procedure is selected to compensate time lags between anemometric variables and gas analyzer measurements. This method determines the time lag that maximizes the covariance of two variables, within a window of plausible time lags (Fan et al., 1990). This has been added in the manuscript.

P5 L11: Add reference for the high frequency correction method. Was this a severe problem?

Answer: The high frequency correction method is referred from Moncrieff et al. (2004), which has been added. The cospectrum analysis shows a rapid decline for the turbulent fluxes at high frequency, indicating the minor effect of small eddy vortex.

P5 L24: It would be useful with a wind rose to illustrate the prevailing wind direction.

Answer: Thanks for your suggestion. The wind rose during daytime and nighttime has been added (Figure 2).

P6 L4: how many data is included in the final data set? How are the data distributed over the different years? Is there any season that has significantly less data than the others?

Answer: After the data quality control, about 66

P6 Section 3: This text is very compact, please separate into paragraphs.

Answer: It has been separated into two paragraphs.

P7, L6: Do you have any more supporting evidence that it is a sea/land breeze circulation? Any land site in the vincinity? Is there any previous studies on this for this site? A single surface observation is in my opinion not enough to support this, it is indicative but not complete shown. As you state on line 11 that strong synoptic (?) west wind during night, this might be explanation for the night wind direction and not land breeze.

Answer: Yes, there is a local meteorological observation site (Dali National Climatic Observatory) near the lake site, with a distance about 4 km between them. The local circulation over Lake Erhai has been analyzed by our another study, which mainly focused on the comparison of atmospheric boundary layer characteristics between pre-monsoon (30 March to 30 April, 2012) and monsoon period (30 June to 31 July, 2012) by conducting sensitivity experiments with the lake-atmosphere coupled model WRF v3.7.1 (Xu et al., 2018). The study has observed the lake breeze circulation during daytime and land breeze circulation during nighttime respectively by analyzing the spatial distribution of horizontal wind. This has been supplemented in the manuscript.

Xu, L., Liu, H., Du, Q., Wang, L., Yang, L., and Sun, J.: Differences of atmospheric boundary layer characteristics between pre-monsoon and monsoon period over the erhai lake. Theor. Appl. Climatol., https://doi.org/10.1007/s00704-018-2386-8, 2018.

P9 L1, "resulted in a lower abledo" compared to what?

Answer: The albedo of Lake Erhai is lower compared to terrestrial land surfaces. This paragraph (the diurnal pattern of radiation components) has been deleted according to another reviewer's opinion.

P9 L8 The Katsaros reference in the reference list is wrong, please use the correct one.

Answer: Sorry for the mistake. This paragraph has been deleted as it's not so interesting to authors according to another reviewer's opinion.

Sections 3.4 to 3.8 please see the general comments

Answer: Thanks for your valuable suggestion. The detailed replies see above.

Figure 8: No data for monsoon and post monsoon for 2014?

Answer: Yes, The instrument failure has resulted in a long data gap for 3 months in this year. This information has been added in the manuscript.

Please also note the supplement to this comment:
https://www.atmos-chem-phys-discuss.net/acp-2018-14/acp-2018-14-AC2-supplement.pdf

─────────────────────────

**Fig. 1.** The figure of radiation sensors's position

Table The correlation between U and Δe, and LE are listed as following:

| Correlation coefficients | 30Min | | | Daily | | | Monthly | | |
|---|---|---|---|---|---|---|---|---|---|
| | pre-monsoon | monsoon | post-monsoon | pre-monsoon | monsoon | post-monsoon | pre-monsoon | monsoon | post-monsoon |
| U and Δe | $0.149^{**}$ | $0.090^{**}$ | $0.244^{**}$ | 0.006 | $0.442^{**}$ | $0.390^{**}$ | -0.482 | $0.505^{*}$ | 0.183 |
| U and LE | $0.502^{**}$ | $0.741^{**}$ | $0.723^{**}$ | $0.634^{**}$ | $0.798^{**}$ | $0.687^{**}$ | $0.791^{*}$ | $0.765^{**}$ | 0.568 |

**Fig. 2.** The table of the correlation coefficiency between U and âŰşe, and LE

**Supplement:**

[revised manuscript text omitted]

---

## Author Response (AR2)

Dear Editor,

Thanks for giving us the opportunity to revise our manuscript again. We have revised the manuscript according to the valuable comments from the anonymous reviewer. All the revisions have been marked with red color in the manuscript. The responses point by point are as following:

**Anonymous Referee**

The authors has essentially revised the manuscript according to my questions and comments. However, one minor but important aspect needs to be addressed. The Dalton and Stanton numbers presented In eq. 1-2 are unfortunately not the ones I was referring to. I am referring to the numbers used in bulk parameterizations of air-sea fluxes see e.g.

Drennan, W. M. (2006), On parameterisations of air‑sea fluxes, in Atmosphere‑Ocean Interactions, vol. 2, edited by W. Perrie, pp. 1–34, WIT Press, Southampton, U. K.

15    A well known algorithm is the COARE algorithm, see e.g.

Fairall, C.W., E.F. Bradley, J.E. Hare, A.A. Grachev, and J.B. Edson, 2003: Bulk Parameterization of Air–Sea Fluxes: Updates and Verification for the COARE Algorithm. J. Climate, 16, 571–591

20    It is more relevant to compare with these versions of the exchange coefficients and e.g. plot them as function of wind speed, compare values during the different periods (pre-, during and post- monsoon), also put them in context, e.g. compare with values from COARE algorithm. In this way you can directly see if your measurements go beyond current state-of-the art knowledge of air-water energy transport (which the COARE algorithm aiming to represent).

The current discussion on Fig 8 and the current versions of the Stanton and Dalton numbers unfortunately doesn't add more understanding of the results. If you argue to keep it, my view is that you need to elaborate more on the interpretation of these results.

*Answer*: Sorry for the misunderstanding about these two parameters (the Dalton and Stanton numbers) at last time. We finally get the real meaning of the Dalton and Stanton numbers, which are referring to heat and moisture bulk transfer coefficients, respectively. We have calculated bulk transfer coefficients during different periods and analyzed the relations between them and wind speed. We also compared our results with other studies on water regimes, as well as the data from COARE. The bulk transfer coefficients over Lake Erhai compared well with values reported in other lake studies. Although there doesn't exist an obvious tendency that the bulk transfer coefficients increase with wind speed in our

study, we think it's not contradictory to the bulk parameterization scheme in COARE (Fairwell et al., 2003), as the magnitude of wind speed over Erhai Lake is relatively lower, corresponding to the transitional range in results from COARE ($< 10$ m s$^{-1}$). The former discussion and figure for last version has been removed. The analysis for the new version and the new Figure 7 has been added in the manuscript.

Typos

P20, line 20, "Lake"a
**Answer:** Thanks for the reminding. The mistake has been corrected.

Figure 2 caption: change to "downward shortwave radiation"
**Answer:** The phrase has been revised.

Figure 9, figure caption, Please double check the caption. Clearly state what upper and lower limits of the large box represent. What does the horizontal line in each box represent? Also how can the 1 and 99% percentiles go beyond the maximum and minimum values represented by the short dash (as I interpret as the whiskers)?
**Answer:** The figure caption has been checked and revised. More details have been added, etc., 
[revised manuscript text omitted]